

# A new aerosol wet removal scheme for the Lagrangian particle model FLEXPART

Henrik Grythe[1,2,3], Nina I. Kristiansen[2], Christine D. Groot Zwaaftink[2], Sabine Eckhardt[2], Johan Ström[1], Peter Tunved[1], Radovan Krejci[1,4], and Andreas Stohl[2]

[1]Department of Applied Environmental Science (ITM), Atmospheric Science Unit, Stockholm University, S 106 91 Stockholm, Sweden
[2]Norwegian Institute for Air Research (NILU), P.O.Box 100, 2027 KJELLER
[3]Finnish Meteorological Institute (FMI), Air Quality Research, Erik Palmenin aukio 1, P.O.Box 503, FI-00101 Helsinki
[4]Division of Atmospheric Sciences, Department of Physics, University of Helsinki, P.O.Box 64 (Gustaf Hällströmin katu 2a) FI-00014 University of Helsinki, Finland

*Correspondence to:* H. Grythe (zhg@nilu.no)

**Abstract.** A new, more physically based wet removal scheme for aerosols has been implemented in the Lagrangian particle dispersion model FLEXPART. It uses three-dimensional cloud water fields from the European Centre for Medium-Range Weather Forecasts (ECMWF) to determine cloud extent and distinguishes between in-cloud and below-cloud scavenging. The in-cloud nucleation scav-

enging differentiates between cloud water phases (liquid, ice or mixed-phase) to allow for aerosol and cloud type specific removal. The impaction scavenging scheme parameterizes below-cloud removal as a function of aerosol size and precipitation type (snow or rain) and intensity.

Sensitivity tests with the new scavenging scheme and comparisons with observational data were conducted for three distinct types of primary aerosols, which pose different challenges for mod-

elling wet scavenging due to their differences in solubility, volatility and size distribution: 1) $^{137}$Cs released during the Fukushima nuclear accident attached mainly to highly soluble sulphate aerosol, 2) black carbon (BC) aerosol, and 3) mineral dust. Calculated e-folding lifetimes of accumulation mode aerosols for these three aerosol types were, 11.7, 16.0, and 31.6 days respectively, when well mixed in the atmosphere. The long lifetimes of mineral dust in particular are primarily a result of

slow in-cloud removal which is the primary removal mechanism.

Calculated e-folding lifetimes in FLEXPART also have a strong size dependence, with the longest lifetimes found for the accumulation-mode aerosols. For example, for dust particles emitted at the surface the lifetimes were 13.8 days for particles with 1 μm diameter and a few hours for 10 μm particles. A strong size dependence in below cloud scavenging, combined with increased dry depo-

sition and gravitational settling, is the primary reason for the shorter lifetimes of the larger particles. The most frequent removal is in-cloud scavenging (85% of all scavenging events) but it occurs primarily in the free troposphere, while below-cloud removal is more frequent below 1000 m (52% of



all events) and can be important for the initial fate of species emitted at the surface, such as those examined here.

For assumed realistic in-cloud removal efficiencies, both BC and sulphate have a slight overestimation of observed atmospheric concentrations (a factor of 1.6 and 1.2 respectively). However, this overestimation is largest close to the sources and thus appears more related to overestimated emissions rather than underestimated removal.

## 1   Introduction

Aerosols are of concern for urban air quality, but also an important part of the climate system. Aerosols enter the atmosphere through primary production (e.g., dust or sea salt emissions) or by nucleation and condensation of gases in the atmosphere, and have both natural and anthropogenic sources (e.g., Seinfeld and Pandis, 2004). Climate is impacted by aerosols both directly by their influence on the radiation budget and indirectly by their influence on cloud optical properties and

precipitation (e.g., Lohmann and Feichter, 2005). Accurate multi-scale modeling of aerosols is difficult due to the variety of processes involved, and thus aerosol-related processes remain the largest source of uncertainty in assessments of anthropogenic radiative forcing (Myhre et al., 2013). Consequently, achieving the best possible representation of the complex processes related to aerosols in models is an important task.

The atmospheric aerosol burdens are controlled by the aerosol sources and rates of removal from the atmosphere. Removal can be through transformation of aerosols (e.g. coagulation to form larger particles; volatilization) or their complete removal from the atmosphere which occurs by dry deposition at the surface and through wet deposition, i.e. removal by precipitation, which is the focus of this study. While dry deposition occurs only at the Earth's surface, wet deposition can remove

aerosols efficiently from the whole troposphere. Clouds can form when moist air is cooled below the saturation point of water vapor (e.g., Rogers and Yau, 1989). Within saturated air, aerosols can act as nuclei for the water vapor to condense upon. The efficiency of aerosols in serving as cloud condensation nuclei (CCN) depends on their size and chemical properties as well as on the ambient conditions. At low temperatures, ice crystals may also form on ambient particles, which then act as

ice nuclei (IN) (Seinfeld and Pandis, 2004). The critical level of relative humidity determining which aerosols are activated to CCN is described by Köhler theory (Köhler, 1928).

When a droplet evaporates completely, non-volatile material is returned back to the atmosphere, but often as transformed CCN or IN with different physicochemical properties compared to the original particles. On the other hand, if the cloud water precipitates to the surface any original CCN or

IN are also removed from the atmosphere. Since each drop of precipitation can account for millions of cloud droplets, nucleation scavenging is the most important mechanism for wet removal (Rogers



and Yau, 1989). Nucleation removal of aerosols within clouds is thought to account for more than 50% of the aerosol mass removal from the atmosphere globally (Textor et al., 2004).

Aerosol particles can also be collected by falling precipitation (Greenfield, 1957; Andronache et al., 2003) through impaction (below-cloud scavenging). The rate at which removal by impaction happens is dependent on the probability of a collision of a falling hydrometeor with an aerosol particle and the efficiency of subsequent collection of the particle by the hydrometeor.

This paper describes and tests a new scheme for aerosol wet removal implemented into the Lagrangian particle dispersion model FLEXPART. It is based on the mechanisms of nucleation removal within the cloud and the impaction removal below the cloud. Section 2 of this paper provides a short description of FLEXPART in general, and introduces the new wet removal scheme. In Section 3, we describe how the new scheme was tested and compared with observations, and section 4 describes the results of these tests. Finally, in Section 5 conclusions are drawn.

## 2   Model description

The Lagrangian particle dispersion model FLEXPART (Stohl et al., 1998; Stohl et al., 2005) computes the transport and turbulent diffusion of atmospheric tracers (e.g., gases or aerosols). The model calculates trajectories based on meteorological input data and can be used from local to global scales. Computational particles follow the flow of the atmosphere resolved in the meteorological input data, with random motions describing parameterized turbulence superimposed on the particles' trajectories. Furthermore, a stochastic particle column redistribution scheme is used to describe convection (Forster et al., 2007). The meteorological data are usually taken from operational analysis or reanalysis products. The reference version of FLEXPART can ingest data from European Centre for Medium-Range Weather Forecasts (ECMWF) or the National Centers of Environmental Prediction (NCEP). Other versions of FLEXPART use e.g. data from the Weather Research and Forecasting (WRF) model (Brioude et al., 2013) or the Norwegian Earth System Model (NorESM) (Cassiani et al., 2016). We base our following discussion on the reference version 10.0 in its configuration for ECMWF data products.

Aerosol removal schemes in Lagrangian models like FLEXPART (Hertel et al., 1995) and indeed other similar models, such as NAME (Jones et al., 2007) or HYSPLIT (Stein et al., 2015; Draxler and Hess, 1998), have remained relatively unchanged since their incorporation in these models in the 1990s (Webster et al., 2014). A reason for this may be the limiting factors that constrain the possible ways of treating aerosol removal within the Lagrangian model framework. A main consideration within this framework is that each transported computational particle is independent of others. Extensions of this concept exist also for FLEXPART (Cassiani et al., 2013), but the reference version of FLEXPART is a purely linear transport model. Within this concept, it is impossible to include aerosol processes which depend on the aerosol concentration (e.g., coagulation or non-linear





chemical reactions). Furthermore, to facilitate consistency between forward and backward runs of
FLEXPART, parameterizations that depend on the age of the aerosol (i.e. time after emission for
primary aerosols) should be avoided as well. This limits the level of sophistication that can be incor-
porated into an aerosol removal scheme. Nevertheless, a realistic treatment of aerosols is possible
even with these limitations.

Each computational particle released in FLEXPART represents an aerosol population with a log-
normal size distribution. While gravitational settling is calculated only for the mass mean diameter
of this aerosol population and applied as an additional vertical velocity component when particles
are advected, dry deposition (for details about the dry deposition in FLEXPART, see Stohl et al.,
2005) is calculated for several weighted bins of the size distribution a particle represents. The parti-
cle mass is then reduced by the dry deposition for the particle as a whole, thus not changing its size
distribution. This simplified treatment of aerosol size distribution can be extended easily by simulat-
ing several different types of particles, each with its own size distribution (or discrete size, if this is
preferred). Removal processes acting differently for the different particle sizes will then also modify
the overall size distribution.

The calculation of wet removal in FLEXPART can be divided in two parts: One regarding the
definition of the location of clouds, cloud water and precipitation, and the other regarding the pa-
rameterization of the physical removal of aerosols and gases during precipitation events. Both parts
have been revised and results will be presented in this paper.

## 2.1 Clouds and precipitation in FLEXPART

For a particle residing in a column with precipitation, it must be determined whether it is located
within the cloud, above the cloud, or below the cloud. Above the cloud, no scavenging occurs;
within the cloud, nucleation scavenging is used; and below the cloud, the impaction scavenging
scheme is activated. The cloud vertical extent can either be derived from three-dimensional ECMWF
fields of specific cloud liquid water content (CLWC) and specific cloud ice water content (CIWC)
or from the summed quantity specific cloud total water content (CTWC = CLWC+CIWC). CTWC
can be calculated by FLEXPART's ECMWF pre-processor to save storage space required for the
FLEXPART input data. Details of how the cloud water is computed by the ECMWF Integrated
Forecast System (IFS) model can be found in Tiedtke (1993); Forbes et al. (2011); Bechtold et al.
(2014) and the processing of these data is described in Stohl et al. (2016). If no cloud water content
data are available in the FLEXPART input files, cloud vertical extent can be diagnosed from the
vertical distribution of relative humidity as in previous versions of FLEXPART (Stohl et al., 2005).
However, this is considered much less accurate.

Multiple layers of clouds may appear both in the relative humidity based parameterization and in
the ECMWF CTWC data. Not all of these cloud layers may be precipitating but, because of lack
of detailed information, in FLEXPART we assume that all levels of clouds contribute to surface



precipitation. An inspection of the ECMWF cloud fields suggests that this assumption is of minor importance as cloud layers with significant gaps in between account for less than 10% of large scale

precipitation events.

Meteorological information in FLEXPART is available only at the resolution of the ECMWF input data. However, a grid cell with precipitation may, in reality, also contain areas without precipitation, and this can reduce the efficiency of aerosol wet scavenging substantially (Sato et al., 2016). The grid surface precipitation intensity ($I_t$) is the sum of the advective precipitation intensity $I_l$ and convective

precipitation intensity $I_c$ from the meteorological input files. To scale this to sub-grid precipitation intensity ($I$) the empirical relationship for the fraction of a grid cell experiencing precipitation ($F$) is maintained from previous versions of FLEXPART, described in Stohl et al. (2005). If a particle is found to be in or below a cloud with precipitation, the scavenging coefficient $\Lambda$ is determined by either the in-cloud or below-cloud scheme described in the following two sections.

**2.2  In-cloud removal in FLEXPART**

The nucleation scavenging in FLEXPART is activated only for particles residing in the precipitating fraction of a grid cell ($F$, see Stohl et al., 2005), and only at altitudes where cloud water is present. For consistency with $I$, the column cloud water is also scaled by the precipitating fraction of the clouds, to get the sub-grid precipitating cloud water ($PCW$):

$$PCW = CTWC \, \frac{F}{cc} \tag{1}$$

Here, $cc$ is the surface cloud cover and so $F/cc$ is the fraction of cloud water in the precipitating part of the cloud.

An important intermediate quantity to determine is the in-cloud removal rate of aerosols due to the removal of cloud water by precipitation, which is given by the washout ratio $I/PCW$. To obtain

accurate values for $I/PCW$, it is important that $I$ and $PCW$ are consistent. Both values are derived from ECMWF data, however, $I$ is derived from accumulated precipitation values (i.e., precipitation accumulated during one ECMWF data output interval, typically 1 or 3 hours), whereas, $PCW$ is an instantaneous quantity, and this can cause small inconsistencies. Furthermore, $I/PCW$ does not take into account the efficacy of turbulent overturning and the replenishment rate of cloud water

from condensing water vapour. The aerosol scavenging coefficient $\Lambda$ (s$^{-1}$) is now given as

$$\Lambda = F_{nuc} \frac{I}{PCW} \, ic_r \tag{2}$$

where $F_{nuc}$, the nucleation efficiency, is the fraction of a given aerosol within a cloud volume that is in the cloud water (see Fig. 1). The constant $ic_r > 1$ is related to the cloud water replenishment rate. It must be noted, however, that $ic_r$ is based on model testing, as the replenishment rate cannot be





determined from the ECMWF output data. It is currently set to a value of 6.1 for the ECMWF cloud water fields. For simulations where in-cloud removal constitutes a large fraction of the removal, i.e. especially soluble accumulation mode aerosols, the value of $ic_r$ has a large impact on overall removal rates.

    In reality, $F_{nuc}$ depends on many different variables such as aerosol size, chemical composition,
surrounding aerosols, temperature and cloud phase and microphysical properties. However, a complete parameterization of $F_{nuc}$ is not possible in FLEXPART because of a lack of information. What can be constrained within FLEXPART is that most aerosols have very different nucleation efficiency for liquid, mixed-phase and ice clouds. We therefore consider the nucleation efficiency $F_{nuc}$ to depend on the efficiency of aerosols to serve as cloud condensation nuclei ($CCN_{eff}$) and ice nuclei
($IN_{eff}$). For ice clouds, $F_{nuc}$ is set equal to $IN_{eff}$, for liquid water clouds, $F_{nuc}$ is set equal to $CCN_{eff}$, and for mixed-phase clouds, we use $\alpha$, the fraction of the cloud water in ice phase shown in Fig. 1 as a black line (see Stohl et al., 2016, for details on calculations of $\alpha$), to interpolate between $F_{nuc}$ and $CCN_{eff}$:

$$F_{nuc} = (1 - \alpha) \, CCN_{eff} + \alpha \, IN_{eff} \qquad (3)$$

There are no unique globally representative values for $CCN_{eff}$ or $IN_{eff}$ because they depend not only on the aerosol particle itself, but vary also with aerosol concentrations and cloud properties (e.g., updraft velocities). Some general considerations can however be made. In a review of measurements conducted at the high alpine station Jungfraujoch, Bukowiecki et al. (2016) showed that $F_{nuc}$ varies significantly with both aerosol size and cloud phase. Henning et al. (2004) found that the fraction of
particles with $d_p > 100$ nm activated in a cloud dropped from 56% in liquid summer clouds to 0.08% in winter ice clouds. The lower ice phase values are attributed to the Bergeron-Findeisen process (Bergeron, 1935; Findeisen, 1938), by which relatively few ice crystals grow at the expense of many more liquid droplets. When the droplets evaporate the non-volatile aerosol content is released back to the atmosphere. This temperature dependent effect is illustrated in Fig. 1, where the partitioning
between cloud water and surrounding air of total aerosol number according to Henning et al. (2004) is shown (magenta dots). Also shown in Fig. 1 are the similar results of Verheggen et al. (2007) (red line) and the BC partitioning (blue line) reported by Cozic et al. (2007). Hence it is generally assumed that for most aerosol particles $CCN_{eff} > IN_{eff}$.

    Gieray et al. (1993) found that the average scavenged fractions in clouds during spring in Cumbria,
U.K., were 0.77 for sulphate and 0.57 for soot in clouds formed in continental air, and 0.62 and 0.44 respectively, for clouds formed in marine air. The time and place for these measurements suggest that these were mainly liquid phase clouds. In other studies (Noone et al., 1992; Gillani et al., 1995; Hallberg et al., 1994), it was found that larger aerosol particles have a higher nucleation efficiency than smaller particles. Such information can be used by FLEXPART users to prescribe appropriate
$CCN_{eff}$ and $IN_{eff}$ values for different particle types and sizes.



### 2.3 Below-cloud removal in FLEXPART

Raindrops and snow flakes fall at approximately terminal velocity through the air (Pruppacher and Klett, 1978) and may scavenge aerosol particles as they collide with them in the ambient air below the cloud base. This below-cloud scavenging process depends both on the probability that the falling hydrometeor collides with an aerosol particle (collision efficiency) and the probability of attachment (coalescence efficiency). Both probabilities together determine the collection efficiency. Whilst Brownian diffusion is the dominant process of attachment for sub-micron particles, inertial impaction is the dominant process for aerosol sizes above 1 μm, though there are large discrepancies between theoretical predictions and observations (e.g., Volken and Schumann, 2007). The collection efficiency is strongly dependent on the sizes of both the falling hydrometeors (and their terminal velocity) and the aerosol particles. It also depends on the precipitation type.

The below-cloud scavenging parameterization in FLEXPART differentiates between rain and snow because especially for large aerosol particles a large difference in scavenging efficiency is found between the two, where snow is more efficient than rain (Kyrö et al., 2009; Paramanov et al., 2011). Of many possible parameterizations for liquid precipitation, the one of Laakso et al. (2003) was chosen, for which all the required information is available in FLEXPART. The parameterization takes into account rain intensity $I$ (used to parameterize droplet size) and the aerosol dry diameter and is based on field measurements over six years in Hyytiälä, Finland. The scavenging coefficient $\lambda$ $(\mathrm{s}^{-1})$ for particles below a cloud is given by

$$log_{10}(\frac{\lambda}{\lambda_0}) = C_*(a + b\,d_p^{-4} + c\,d_p^{-3} + d\,d_p^{-2} + e\,d_p^{-1} + f(\frac{I}{I_0})^{0.5}) \tag{4}$$

where $d_p = log_{10}\frac{D_p}{D_{p0}}$, $\lambda_0$=1 $\mathrm{s}^{-1}$, $I_0$= 1 mm hr$^{-1}$, and $D_{p0} = 1$ m. Coefficients for factors $a - f$ are given in Table 1. While originally intended for particles of size 10-510 nm, the parameterization by Laakso et al. (2003) is one of few parameterizations that takes into account data for larger aerosol particles up to 10 μm diameter, and should thus provide reasonable results also for these larger particles.

For snow scavenging, we use a parameterization reported by Kyrö et al. (2009), which was also derived from Hyytiälä data, but during snowfall. It is fitted with the same function as given by Eq. 4 but with coefficients derived for snow and also given in Table 1. A threshold surface (2 m) temperature of 0°C is used to distinguish between rain and snow, unless rain and snow precipitation intensity is read directly into the model from ECMWF analysis data. The Kyrö function is independent of precipitation intensity or type of falling snow as is common for snow scavenging parameterizations (see e.g., Paramanov et al., 2011; Zhang et al., 2013). The shape of the snow crystals is very important for the scavenging efficiency, but cannot be derived from the ECMWF data. This aspect is thus ignored, and the Kyrö function is averaged over many different types of snow crystal shapes instead.





Fig. 2 shows the below-cloud scavenging parameterizations for rain and for snow for different
precipitation rates and compares them with the old parameterization used in FLEXPART, which was
based on Hertel et al. (1995). The aerosol removal rate is increased relative to previous versions
of FLEXPART for almost all precipitation rates. Aerosol chemical properties may also influence
the below-cloud scavenging coefficient. In FLEXPART, this influence can – to some extent – be

accounted for by setting the parameters $C_{rain}$ and $C_{snow}$ ($C_*$ in Eq. 4), which are scalars used
to scale the collection efficiency for rain and snow, to values different from 1. For example, with
$C_{rain} = 0$ ($C_{snow} = 0$), no below-cloud scavenging for rain (snow) would occur in FLEXPART.

As parameterizations by both Laakso et al. (2003) and Kyrö et al. (2009) are based on bulk aerosol
there may be differentiating factors for certain aerosol types, though very little specific evidence of

this exists (Zhang et al., 2013). Comparisons with other impaction scavenging parameterizations
(see e.g., Zikova and Zdimal, 2016) for rain show that the Laakso et al. (2003) scavenging values
are on the middle to low side of existing parameterizations and that differences between different
parameterizations cover at least one order of magnitude. Choosing values for $C_{rain}$ and $C_{snow}$
between 0.1-10 should cover this uncertainty range.

**3   Model simulations**

Three different global model experiments were set up to test the new scavenging parameterizations
for different types of primary aerosols: BC, mineral dust and sulphate. The main purpose of these
experiments is to explore the performance of simulations that cover a broad range of aerosol particle
types and sizes, evaluate simulated atmospheric concentrations against observations, and calculate

e-folding lifetimes.

**3.1   Mineral dust**

Mineral dust arguably constitutes the largest mass of aerosols in the atmosphere. Dust particles
span a wide range of sizes and can be found far from their source (Reid et al., 2003). Small dust
particles have been found to mix somewhat with volatile aerosol components but particles larger

than 0.5 μm are inert in the atmosphere (Weinzierl et al., 2006). Mineral dust is thus well suited
to model with FLEXPART. Model experiments were set up to examine the role of impaction and
nucleation scavenging as well as dry deposition for different sizes of mineral dust.

Emission of mineral dust was calculated based on a module presented by Groot Zwaaftink et
al. (2016). In short, dust emission was initiated from bare land when friction velocity exceeded a

threshold value for initiation of saltation, depending on soil properties and soil moisture content.
The soil fraction available for erosion was determined from land cover data (GLCNMO version 2,
Tateishi et al., 2014) based on MODIS images. Vertical fluxes of mineral dust were derived according
to Marticorena and Bergametti (1995). Particles were subsequently released in FLEXPART over a





layer of 300 m height, at a 0.5 degree resolution in 6-hourly time steps. We assumed a particle size

distribution in ten particle size bins, varying between 0.2 and 18.2 μm, as suggested by Kok (2011).
FLEXPART simulations were run in forward mode for the year 2010.

### 3.2 Radionuclide tracers attached to sulphate aerosols

An evaluation of modelled aerosol lifetimes was recently performed by Kristiansen et al. (2015) who
made use of measurements of radioactive isotopes released during the Fukushima Dai-Ichi nuclear

power plant (FD-NPP) accident in March 2011. The radionuclide cesium-137 ($^{137}$Cs) was released
in large quantities during the accident and measurements suggested that they mainly attached to the
ambient accumulation-mode sulphate aerosols (Kaneyasu et al., 2012). Another radionuclide, the no-
ble gas xenon-133 ($^{133}$Xe) was also released during the accident and can serve as a passive transport
tracer. Both radioactive isotopes were transported and measured across the Northern Hemisphere

for more than three months after their release, providing a unique constraint on modelled aerosol
lifetimes (Kristiansen et al., 2015).

We have used measurements of the aerosol-bound $^{137}$Cs and the noble gas isotope $^{133}$Xe from
March to June 2011 at 11 different measurement stations of the Comprehensive Nuclear-Test-Ban
Treaty Organization (CTBTO) network (see Figure 1 of Kristiansen et al., 2015). All measured

radionuclide concentrations were corrected for their radioactive decay and converted to activity per
cubic meter for comparison with the model data. Detailed descriptions of these measurements and
how they can be used to determine aerosol e-folding lifetimes were provided by Kristiansen et al.
(2012, 2015).

Over the 46 days of measurements (starting 14 days after the initial emission) used to evaluate

e-folding times of $^{137}$Cs (and, implicitly, the accumulation mode sulphate aerosol to which it at-
tached), Kristiansen et al. (2015) found FLEXPART concentrations to decrease by three orders of
magnitude more than the measurements. The decrease started from an initial overestimation of the
$^{137}$Cs concentrations but later the concentrations were underestimated at all but one CTBTO stations.
Consequently, a too short e-folding lifetime of 5.8 days was calculated for FLEXPART as compared

to 14.3 days derived from the measurements. In this paper, we repeat the simulations of Kristiansen
et al. (2015) but with the new removal scheme for aerosols.

### 3.3 Black carbon

FLEXPART has been used in several recent studies to model BC with a focus on the Arctic (Stohl
et al., 2013; Yttri et al., 2014; Eckhardt et al., 2015). All these studies used a FLEXPART version

where the in-cloud scavenging efficiency of the reference FLEXPART version had been reduced by
one order of magnitude. This has produced realistic concentrations for the Arctic. In this study, we
tested the new scheme against measurements at Arctic and mid-latitude stations to assess how well
BC concentrations are captured.



For BC, simulations were made both in forward and backward mode, and results were compared
to test model consistency. When run in backward mode, FLEXPART output is a gridded emission
sensitivity that can be coupled with emission fluxes to obtain the concentrations at the release point.
For all simulations, concentrations obtained by forward and backward simulations by FLEXPART
differ only due to statistical noise.

Emissions used for BC simulations were ECLIPSE v4.0 (Stohl et al., 2015) available through
the website http://eclipse.nilu.no. Added to these are shipping emissions from AEROCOM (Den-
tener et al., 2006) and GFEDv3.1 emissions for forest and savannah fires (Randerson et al., 2013;
Van der Werf et al., 2006), all resolved monthly and on a 0.5°x0.5°grid. European measurements
of aerosol absorption were collected from the Database for Atmospheric Composition Research
(EBAS) database with the aim of using data from stations with similar particle soot absorption pho-
tometer (PSAP) instruments. The stations were selected to represent different environments, rang-
ing from locations close to pollution sources in Central Europe to remote locations in the Arc-
tic. We chose the sites Melpitz (MEL, 51.32° N 12.56° E) in Germany which is surrounded by
strong BC sources, Pallas (PAL, 67.80° N 27.16° E) in Finland and Southern Great Plains (SGP,
36.50° N 98° W) in the US at intermediate distances from the sources, and Zeppelin (ZEP, 78.93° N
N, 11.92° E), Barrow (BAR, 71.30° N, 156.76° W) and Alert (ALT, 82.50° N, 62.34° W) as remote
sites.

PSAPs measure the particle light absorption coefficient. Conversion of this coefficient to equiva-
lent BC (eBC) mass concentrations is not straightforward and requires certain assumptions (Petzold
et al., 2013), leading to site-specific uncertainties on the order of a factor of two. We have used con-
version factors of $6.50\,\mathrm{m^2 g^{-1}}$ for PAL and $5.50\,\mathrm{m^2 g^{-1}}$ for ZEP, where site-specific information was
available and $10\,\mathrm{m^2 g^{-1}}$ for MEL, ALT, BAR and SGP. For ALT and BRW a gap with more than a
month of missing data for 2007 was filled with climatological values of all available data after year
2000. For PAL only climatological observations were used.

## 4   Results

### 4.1   Wet scavenging event statistics

To explore how frequent in-cloud and below-cloud scavenging events are and where they occur, we
used a three-months (December 2006 to February 2007) global ECMWF data set (1°x1°with 92
vertical layers) and classified each grid cell as being either outside a cloud, in a cloud, below a cloud
or above a cloud. The vertical extent of each layer increases with altitude, which emphasises lower
altitudes when a raw count of events is done, so for a more realistic representation the numbers pre-
sented here are weighted by the mass of each model layer (using a standard atmosphere). Convective
and large scale precipitation events were differentiated using surface precipitation and for each event
classified as the larger of the two.



Cloud top heights and the frequency of scavenging events are shown in Fig. 3, both using the
ECMWF cloud water information (blue) and the cloud parameterization based on relative humidity
(red). Close to the equator, the precipitating clouds from ECMWF have on average high cloud tops,
often extending all the way to the tropopause. For the period examined, more than 96% of the in-
cloud removal events in the tropical band (15 °S-15 °N) are convective. For the 15-60° latitude range
the cloud tops are markedly lower and the frequency of convective removal events drops markedly
to 46% which is a result of both more stratiform clouds and fewer and lower convective clouds.
This can be seen in the left panel of Fig. 3 as an extension of the 25-75% percentile range, which
indicates that there are both low stratiform and high convective cloud tops. The fraction of large
scale in-cloud events in this area is 46%. Poleward of 60°, stratiform precipitation dominates with
76% of all events.

Globally, in-cloud scavenging accounts for 85% (91% above 1000 m) of the aerosol wet removal
events, of which 57% occur in convective clouds. The global fraction of in-cloud (solid line), below-
cloud (dashed) and total (dotted) removal events as a function of altitude is shown in Fig. 3 (right).
For the ECMWF defined clouds (blue) there are very few below-cloud scavenging events above
1000 m. There is however a slight increase in the frequency of such events around 5000 m, which is
due to multiple layers of clouds. In the instances where precipitation was predominantly large scale
(21%), at altitudes above 5000 m, in reality most clouds are likely non-precipitating cirrus clouds,
and the ECMWF precipitation is actually originating from lower cloud layers. This could also be
related to both convective and large scale clouds in the same grid, but without information about the
three-dimensional distribution of hydrometeors, a correct diagnosis is not possible and many of the
high-altitude below-cloud scavenging events are probably not real. However, in total this accounts
for only 4% of all below-cloud scavenging events.

The water phase of clouds influences the removal efficiency for aerosols that are inefficient IN
but efficient CCN (or vice versa). The phase partitioning is temperature dependent and varies with
season, latitude and altitude. For the three months examined, globally 16% of the in-cloud removal
events were liquid only, 7% were ice only, whereas the remaining 77% were defined as mixed-phase
cloud removal events.

In previous versions of FLEXPART, clouds were parameterized using relative humidity. As can
be seen in Fig. 3, this leads to several differences in the distribution of scavenging events from the
ice and liquid water based cloud distribution. For instance, the high frequency of clouds extending
all the way to the surface seems unrealistic, and often no clouds could be found in a grid cell with
precipitation (not shown). Altogether, in the new scheme the cloud distribution is more consistent
with the precipitation data and thus it produces a more realistic distribution of scavenging events.
with 52% of the removal below 1000 m as below-cloud.

While Fig. 3 shows the global distribution of scavenging events, the actual relative probability
of in-cloud versus below-cloud scavenging events versus dry deposition events for a given particle



depends on the distribution of the aerosol. To illustrate this, we released a pulse of 1 million particles representing dust of five different sizes (see Table 2 ) at 10 m.a.g.l over Central Europe on 14 April 2007. Fig. 4 shows the relative frequency of the different removal events for these aerosol particles as a function of time after the release. For the purpose of clearer illustration, we show a polynomial

fit through the daily total number of events of each removal type. Initially, below-cloud scavenging and dry deposition are the most frequent removal types. Exact numbers at the beginning will vary depending on the location and time of the release. However, as particles are transported to higher altitudes, the relative frequency of in-cloud removal events increases, exceeding that of the other event types from day 4. On day 7 after the emission pulse, the relative frequencies are already similar

to the global distribution of scavenging events in the troposphere, where below-cloud scavenging accounts for only 15% and dry deposition for only 3% of the number of events. Notice that in terms of aerosol mass removed, the importance of below-cloud scavenging and dry deposition will decrease even more quickly because the mass of particles remaining in the lower troposphere will also decrease rapidly. This effect has been discussed in Cassiani et al. (2013). The time dependence

of scavenging is an important feature as most primary aerosols are emitted at or near the surface. Figure 4 also shows that, despite the global dominance of in-cloud scavenging events, below-cloud scavenging or dry deposition may be most important, at least for aerosol types for which these removal mechanisms are efficient. The dependence in the efficiency and nature of scavenging also means that aerosol lifetimes are different for fresh and aged aerosols, as discussed in Kristiansen et

al. (2012, 2015).

### 4.2 Mineral dust

Since the below-cloud scavenging scheme has a strong size dependency, an important goal for our mineral dust simulations was to investigate the differences in lifetime for aerosol particles with a large range of different sizes. Also, mineral dust particles are ineffective CCN (e.g., Mahowald et al.,

2014) and, therefore, below-cloud scavenging is very important for dust. To investigate the sensitivity of dust scavenging to various components of the scavenging scheme, we performed simulations for a range of parameter settings.

The resulting lifetimes ($\tau_F$) are shown in Table 2. Lifetimes were calculated as the times when the dust mass has decreased to 1/e of the emitted mass. Values of $\tau_F$ are equivalent to e-folding times if

the removal rate is constant. While this is not the case - as shown in the previous section -, it allows a simplified lifetime calculation and is sufficient for our purpose of investigating the systematic dependence of lifetime on particle size and choice of scavenging parameters. It also emphasizes the initial phase of removal when most of the emitted mass is lost.

The accumulation mode particles in the 0.2-0.6 μm size range are located in the minimum of

both impaction efficiency (Fig. 2) and dry deposition velocity. Consequently, and especially since dust particles are also inefficient CCNs, they have very long lifetimes. With the standard parameter



settings in FLEXPART for dust ($C_{snow} = C_{rain} = 1$; $CCN_{eff} = 0.15$; $IN_{eff} = 0.02$, highlighted in green in Table 2), the lifetime of accumulation mode-sized (200 nm) dust is almost 32 days. Even though dust particles are inefficient as CCN, wet removal dominates the total removal for the two smaller reported size bins and nucleation scavenging in liquid water clouds is the dominant removal process. Only if $CCN_{eff}$ is decreased further by one order of magnitude, its importance is diminished and the lifetime increases to >50 days.

The loss of particles of size 2.2 $\mu m$ is more strongly affected by dry deposition, but still dominated by wet removal. Impaction scavenging is also about four times more efficient for aerosols of this size than for 200 nm particles, and thus has a large impact on the atmospheric lifetime. This is important especially close to the sources, when the aerosols are predominantly in the lower troposphere where below-cloud removal occurs most frequently. Consequently, the lifetime $\tau_F$, 11.6 days, is substantially shorter than for the 200 nm particles. There is also a strong sensitivity to the choice of the $C_{snow}$ value for scavenging due to ice, which is probably related to the strong size dependence of the Kyrö et al. (2009) scheme.

For the even larger particles shown in Table 1, gravitational settling and dry deposition take over as the most important removal mechanisms and thus very little effect is seen from altering the wet removal parameters. For the 6.2 μm particles, reducing all wet removal parameters by one order of magnitude, only increases the simulated lifetime by 20%, compared to the 450% increase in lifetime for the accumulation mode particles. For the 18.2 μm particles, wet scavenging has virtually no impact on the lifetime, which is entirely controlled by dry removal.

A multi-year study of mineral dust, using FLEXPART with the same removal as here (Groot Zwaaftink et al., 2016)[1] found very good correlation between observations and model concentrations using a global network of observations positioned at various distances from major source regions. While the 32-day lifetime $\tau_F$ obtained for the 200 nm particles seems long, the emission to column burden estimate of lifetime for the full dust size distribution is 4.3 days, which is even on the low side of commonly reported estimates (e.g., Zender et al., 2004). Notice that the mass fraction of dust aerosols with diameter $< 1$ μm is very low in our emission scheme (Kok , 2011).

### 4.3 Radionuclide tracers representative of sulphate aerosols

The FLEXPART model set-up for simulating the aerosol-bound cesium transport after the Fukushima accident was the same as in Kristiansen et al. (2015), except for the updates in the cloud and wet scavenging schemes described in this paper. Furthermore, Kristiansen et al. (2015) used only one aerosol size mode, with d=0.4 μm. Here, a more realistic aerosol size distribution was used, and compared to the measurements of $^{137}$Cs surface activity by Kaneyasu et al. (2012). For these simulations, the mass was emitted in six different size bins (Table 3) ranging from 0.4 μm – 6.2 μm. The

---

[1]The values stated in Groot Zwaaftink et al. (2016), have been changed in Table 2 to correspond to the settings of $ic_r = 6.2$ used here.





size bins with logarithmic mean diameters of [0.4 ,0.65,1,2.2, 4, and 6.2] μm received 1, 2, 10, 40, 32, and 15 % of the emitted mass. The resulting relative aerosol surface size distribution is shown in Fig. 5b at the time of the release (green) and for an aged distribution after 40 days (cyan) together with the measured $^{137}$Cs aerosol surface activity size distribution (red) of Kaneyasu et al. (2012).

It is worth noting that Kaneyasu et al. (2012) started their measurements 47 days after the largest emission but probably sampled mainly $^{137}$Cs from small later releases. The measured size distribution of $^{137}$Cs is bimodal with peaks around 1 μm and 0.02 μm. The larger peak at 1 μm fits well with both the aged and the released size distribution in FLEXPART. While the initial release included a significant fraction of particles larger than 1 μm (52% by mass and 7% by aerosol number), their

fraction is reduced considerably by day 40 (3% by mass and <0.1% by number). The smaller mode around 0.02 μm is not represented in the model but it accounts for only 5-6% of the total mass.

For evaluating the modelled aerosol lifetimes in the same way as Kristiansen et al. (2015), we calculate the ratio of the aerosol ($^{137}$Cs) to the passive tracer ($^{133}$Xe) at each measurement station shown in Fig. 5a. The ratios decrease with time due to removal of aerosols. We further calculate

the daily median ratios (median concentration for each day over all stations), and fit an exponential decay model (grey lines in Fig. 5c) to these daily ratios. The fit is done over days 15 to 65 after the start of emissions, for which sufficient measurement data exist (see Kristiansen et al., 2015, for details). This excludes the initial phase of removal (as shown in Fig. 4) and thus emphasizes the role of in-cloud scavenging. We therefore use the e-folding time of the exponential decay model as an

estimate for the aerosol lifetime ($\tau_e$).

The e-folding lifetime estimate obtained by Kristiansen et al. (2015) for the previous version of FLEXPART was 5.8 days, indicating a too quick removal of the aerosols compared to the measurement-derived $\tau_e$ value of 14.3 days. However, there was only a slight underestimation of the atmospheric concentrations, partly explained by an initial overestimation. The new scavenging scheme produces

a longer e-folding lifetime of 10.0 days (Fig. 5c). The longer lifetime is mainly due to slower in-cloud scavenging and a broader range of aerosol particle sizes emitted, which have different removal efficiencies. Both the below-cloud scavenging as well as the dry deposition are size-dependent. This also explains the shift towards smaller particle sizes from the initial distribution to the aged distribution in Fig. 5b.

The e-folding times calculated individually for the different size bins are reported in Table 3. Simulation #1 in the top row (green) show the results with scavenging parameters set to values believed to be valid for sulfate, which are also used in the simulation shown in Fig. 5. The e-folding lifetimes range from 10.8 days for the 0.4 μm size bin, to 5.4 days for the 4 μm bin. Even the smallest two aerosol size bins have a shorter e-folding lifetime than what is derived from the CTBTO measure-

ments. For the largest size bin, concentrations after 15 days were too low for a robust estimate of lifetime.



The second column in Table 3 for each aerosol size bin reports the ratio of modeled to observed concentrations averaged over the whole period, assuming that all $^{137}$Cs was attached to aerosols of that size bin. Assuming that $^{137}$Cs attached exclusively to particles smaller than 1 μm (first two size bins), which have the most realistic lifetimes compared to the observation-derived lifetime, leads to a large overestimate of the observed concentrations (ratios of 18.7 and 11). This might to some extent be due to an overestimate of the emissions used here, by Stohl et al. (2012). Indeed, other authors (e.g. Morino et al., 2011) have found smaller emissions, but the source term uncertainty of about a factor of two cannot alone explain the overestimates by the smaller modes. Assuming that all $^{137}$Cs attached to particles larger than 2.2 μm, on the other hand, leads to underestimates of both the concentrations and the lifetimes compared to the observations.

From the differences between the simulations for different aerosol sizes, it is also possible to investigate the relative importance of different removal mechanisms for the different aerosol sizes. Furthermore, several different in-cloud parameters $IN_{eff}$ and $CCN_{eff}$ were tested. In simulations #2 and #3 in Table 3, $IN_{eff}$ and $CCN_{eff}$ were reduced to values of 0.4 and 0.15, respectively. In simulations #4 and #5, in-cloud and below-cloud scavenging were separately turned off completely. For these simulations, only one aerosol size was used. Comparison of the lifetimes and ratio of these simulations with the original $^{137}$Cs simulation #1 (Table 3) shows that for submicron particles the governing removal process is in-cloud scavenging. For particles in the range ∼50-800 nm, dry deposition is slow and also the below cloud removal in FLEXPART is not very efficient, which leaves in-cloud scavenging to control the lifetime. This is apparent from how changes in removal efficiency influence the model values and lifetimes differently for different aerosol sizes. When $CCN_{eff}$ and $IN_{eff}$ are reduced by 60% to 0.4 in simulation #2, the atmospheric burden is increased by a factor of 5 for 0.4 μm particles. The lifetime however, only changes from 11.7 to 17.9 days, i.e. by a factor of ∼1.6. For the four larger aerosol size bins much smaller changes are found between #1 and #2 in concentration, lifetime and ratio, due to the less dominant role of in-cloud scavenging for these particles.

The measurement data during the first 15 days after the start of the emissions are insufficient to derive an aerosol lifetime. However, for the model simulation #1, the intermittent e-folding time for the full size distribution of $^{137}$Cs during the first 15 days is 6.1 days, compared to the 10 days found over the 45 day period in Table 3. This is due to the reduction of below-cloud scavenging and dry deposition events (shown in Fig. 4) combined with a reduction of in-cloud scavenging as well, as after 15 days a large and increasing fraction of the left-over aerosol particles reside above the cloud tops. As particles with more efficient removal are lost, the lifetime is more and more influenced by the longer-lived particles over time and thus the model e-folding lifetime estimate increase with time. This effect applies in FLEXPART only when the aerosol size distribution consist of more than one aerosol type (i.e. modal size or different removal parameters).



In Fig. 5d the mean model / observed concentration ratios at the different stations are plotted against latitude. A prominent feature of FLEXPART and indeed most models used by Kristiansen

et al. (2015) is a tendency to overpredict concentrations at low latitudes and underpredict concentrations at high latitudes. This tendency is also present with the new removal scheme, where model / observation ratios decrease with latitude. A plausible explanation of this might be that in-cloud scavenging in ice clouds is too effective. However, sensitivity simulations where $IN_{eff}$ only was reduced (not shown) revealed that this change had little effect, and the latitudinal bias was not reduced

significantly. The probable cause of this is the high proportion of mixed phase clouds (77%) which reduces the impact of the latitudinal dependence of the frequency of ice-phase clouds. It may also be relevant that the clouds have on average higher cloud tops near the equator, so that temperature and thus the mixing state of clouds does not have a strong enough latitudinal dependence in the Northern Hemisphere at the time of this simulation (March-May).

**4.4 Black carbon**

BC has been proven notoriously difficult to model accurately. For example, Arctic seasonal variations and Arctic haze periods are not captured well in most models (Lee et al., 2013). Some of this can be accredited to BC aerosol undergoing stages of transformation after its release to the atmosphere from a hydrophobic to a hydrophilic state (e.g., Bond et al., 2013). The aerosol ageing

processes that would influence in-cloud scavenging are not readily included in FLEXPART and the constant removal parameters cannot account for this transformation. Therefore, several aerosol parameter combinations were tested with FLEXPART both in backward and forward mode. There are observations that urban BC is transformed very quickly into particles with aged, hydrophylic characteristics (Wittbom et al., 2014). Therefore, a representation resembling physical properties of aged

BC (BC #1 in Table 4) was selected as our reference set-up for BC. Our assumptions regarding the values of $CCN_{eff}$ and $IN_{eff}$ were based on the findings of Cozic et al. (2007) that BC is much more efficiently removed in liquid water clouds than in ice clouds. Noone et al. (1992) showed that aerosol composed of mainly elemental carbon had the highest fraction of non activated particles. A size distribution with a modal mean diameter of 150 nm was assumed.

In addition to our simulations for our reference BC species, seven other simulations were performed to test the sensitivity of model results at different latitudes, altitudes and times of the year to changes in the parameters describing the different removal mechanisms. For this, parameter settings were varied within ranges thought to be suitable for BC. Table 4 summarizes the parameter choices for these simulations.

Column burdens and vertical distribution of the eight simulations are shown in Fig. 6. The concentrations are FLEXPART output from five vertical layers with upper borders of 100, 1000, 5000, 10000 and 50000 m. The BC column burdens (shown with white lines in Fig. 6 on the right hand side y-axis) are overall somewhat high when compared to other studies (e.g., Lee et al., 2013; Eckhardt



et al., 2015), with the exception of simulation #4, which has strongly enhanced in-cloud removal.
The dashed black line shown in all the panels is the concentration of the reference simulation (#1).

All simulations produce a quite similar latitudinal distribution. The strongest sources of BC are at mid latitudes and most of BC at high altitudes is also found in this region for all simulations. Thus, the highest column burdens are found near 35° N in all simulations. The two simulations with reduced in-cloud scavenging (#2 and #8), have the highest column burdens. While increasing $C_{rain}$
by a factor of 10 (simulation #5) reduces the burden significantly, a similar, but an even stronger effect can be achieved with a reduced aerosol size (simulation #3), as smaller particles have higher dry deposition velocities. This shows that in the absence of efficient wet removal, dry deposition can be important as well. Though it generally accounts for less than 10% of total removal in our simulations for particles with $d$<1 μm , in simulation #3 it accounts for 48% of the removal. Only
simulations #6 and #7, which have phase dependent changes to removal parameters produce burdens with a substantially different dependence on latitude when compared to simulation #1.

Annual average calculated BC concentrations in the surface layer (0-100 m) in the northern hemisphere are shown in Fig. 7 for the reference simulation (top left) and as differences from this reference for the other seven simulations. Overall, there are only small differences between the various
model runs in the major BC source regions, where the concentrations are strongly influenced by local emissions. Further away from the source regions, differences in removal have a stronger effect. Simulation #4, with enhanced in-cloud scavenging in both liquid and ice clouds, stands out with very low concentrations in the Arctic and other remote regions. The remaining simulations have concentrations within ±50%. It is worth noting that there are quite a few distinct geographical features in
Fig. 7. For example, for simulations #2, and #8, where in-cloud removal is reduced, modelled surface concentrations are increased in remote areas like the Arctic. Turning off the below-cloud removal by snow (simulation #6), however, only has a small effect north of 60° N.

The monthly measured (black) and modelled (blue; simulation #1 in Table 4) BC concentrations at six measurement stations are shown in Fig. 8. The station locations are marked in Fig. 6 and are
at different distances from major source areas. The aerosols measured at the different stations thus have very different ages. For simulation #1, at Melpitz the mean mass weighted FLEXPART aerosol age is 1.3 days, at Pallas it is 3.8 days and at Zeppelin it is 7.7 days. The age is defined as the time it takes for the aerosol to reach the station after its emission. The aerosol age depends not only on the transport, but also on the removal between emission and observation.
Increased removal efficiency would, on average, reduce aged BC more than fresh BC, resulting in a less aged aerosol population. Systematic differences in model bias for stations close to and stations far away from source regions can thus allow to separate errors in emissions versus errors in simulated aerosol lifetimes. In Table 4 the median modelled concentrations at the six stations are reported for all the sensitivity simulations. Seven of the eight simulations overestimate the concentrations at
Melpitz by a factor of almost 2, especially in summer (Fig. 8). This suggests that local emissions



around Melpitz are too high, as changes in the removal parametrisation have little effect on the concentrations (Tab. 4).

Moving away from the source regions, stations Southern Great Plains and Pallas have model concentrations close to the observed average for all the simulations except for simulation #4 which
underpredicts the concentration at these two stations by a factor of 2.1 and 8, respectively. Annual mean BC concentrations at the three Arctic stations Alert, Barrow and Zeppelin are underpredicted by the model (mainly due to very low simulated summertime concentrations, see Fig. 8). This alone would indicate a too fast removal and thus a too short BC lifetime. However, indicative of total global removal rates, the column burden is, also for the Arctic, on the high side of most current
model estimates (Breider et al., 2014) and therefore also burden / emission estimates of the BC lifetime of 9.0 days is higher than in many other models (Samset et al., 2014).

Observations at all stations except Southern Great Plains have a seasonal cycle, with lowest concentrations during summer and higher concentrations during winter. The Southern Great Plains station has a somewhat different seasonality than the other stations, with a peak in autumn, and this is
quite well captured by the model. The four higher-latitude stations all show a pronounced winter / spring peak, which is well reproduced by the model.

In Fig 9 (bottom panel) a comparison between the observations and model simulations #1 and #7 is shown as a 48 hr moving average. With a Pearson's squared correlation coefficient of $r^2$=0.44, simulation #1 captures nearly half of the variability of the observations with generally higher con-
centrations during December to May, and large peaks in the observations in January and December. The difference between simulation #1 and #7 is about 10% and only visible in winter, where the effect of removal differences between the two setups is largest. The difference between the other simulations (not shown) is rather constant throughout the year and thus can be gleaned from the average bias (Table 4).
FLEXPART aerosol age at Zeppelin was also used to examine the role of the removal processes in the variability. Tunved et al. (2013) showed observed concentrations of aerosol submicron mass had a strong dependence on trajectory accumulated precipitation. Shown in the top panel in Fig. 9 is the mean model age corresponding to six-hourly observations. Also a smoothed 48 hr fit is shown in red. Depending on the season, the youngest BC aerosols are found in combination with either the
highest observed concentrations (winter) or very low concentrations (summer) and have two different explanations. The high aerosol episodes with low age in winter (e.g., peak on 15 January) are related to fast transport from the Yamal Peninsula and the Kara Sea. In this area there are large emissions from the gas and oil industry (Stohl et al., 2013), and if these emissions are transported quickly and nearly without removal to Zeppelin, concentrations there increase strongly. In summer, there is
a persistent background of relatively low aerosol concentrations. Occasionally, this background is reduced further by scavenging events occurring close to the station. This removal only leaves BC



from the small local sources of BC on Svalbard, leading to both a low age and low concentrations of the simulated BC.

## 5   Summary and conclusions

This paper has presented the new FLEXPART aerosol wet scavenging scheme. Firstly, a more realistic distribution of clouds was achieved by incorporating three-dimensional cloud information from ECMWF. Secondly, a new parameterization of wet removal within and below clouds was introduced, considering also the water phase of the clouds and the precipitation type.

Reading of cloud liquid and ice water data from stored ECMWF data leads to fewer inconsisten-
cies with the ECMWF precipitation data than using the old relative humidity-based cloud scheme, and is an important improvement of FLEXPART. Using the ECMWF cloud water data, we diagnosed the frequency of different types of removal events, and we found a dominance of in-cloud scavenging events (91% of all events) above $1000\,\mathrm{m}$. At lower altitudes than $1000\,\mathrm{m}$, below-cloud scavenging events are slightly more important (52% of all events) than in-cloud scavenging events.

We performed model simulations for three different types of aerosols (mineral dust, $^{137}$Cs attached to sulfate, and BC), to test different aspects of the removal scheme. For each of these aerosol types, we performed sensitivity simulations to explore the size dependence of the aerosol removal, to determine atmospheric e-folding times, and to investigate the water phase dependency of the aerosol removal scheme. We also compared simulation results to observations of $^{137}$Cs and BC.

For both mineral dust and $^{137}$Cs simulations, the aerosol lifetime had a maximum in the accumulation mode of 31.8 and 11.7 days, respectively. For the BC particles, which are also in the accumulation mode, an e-folding lifetime of 16 days was found. These lifetimes are long compared to lifetimes typically reported in the literature. However, this can be explained by differences in the definition of lifetime (see discussion in Kristiansen et al., 2015). For instance, estimating the life-
time by dividing the aerosol burden with its emission rate - a common definition of lifetime used by global aerosol modelers - results in a BC lifetime of $9\,\mathrm{days}$. This is quite comparable to lifetime values reported for BC in the literature, though perhaps still somewhat longer than in most models (Samset et al., 2014; Cape et al., 2012; Ogren and Charlson, 1983).

In our scheme, the lifetime of accumulation mode particles is controlled mainly by in-cloud re-
moval, as dry deposition and below-cloud scavenging are inefficient for these particle sizes. Therefore, the longer e-folding lifetime is due mainly to transport of aerosols above clouds, where they cannot be scavenged. This process is less important for the burden/emission lifetime estimate, which depends mostly on the particle removal rate in the first few days after the emission.

Simulations for the accumulation mode particles with FLEXPART are highly sensitive to the
choice of $CCN_{eff}$ and $IN_{eff}$ values, which describe the particles' efficiency to serve as cloud condensation and ice nuclei. On the other hand, for particles larger than $1\,\mu\mathrm{m}$, both below-cloud scav-





enging and dry deposition have a strong impact on the lifetime. Consequently, these larger aerosols all have much shorter e-folding times. CCN (and IN) efficiency has also been shown to increase with aerosol particle size, thus contributing to the shorter lifetimes of particles larger than 1 μm. However,

as their lifetime is mostly controlled by dry deposition and below-cloud scavenging, the choice of values for $IN_{eff}$ and $CCN_{eff}$ is not particularly critical for super-micronic particles.

There are large uncertainties tied to the efficiency of impaction scavenging. Nevertheless, the chosen schemes of Laakso et al. (2003) and Kyrö et al. (2009) capture the overall size dependence predicted by impaction theory. For BC, for which the removal by snow is generally more efficient

(especially at low precipitation intensities), taking into account the precipitation water phase leads to relatively stronger removal of BC at high latitudes and so enhances the underestimation of BC concentrations at the Arctic stations. This effect is however small compared to the aerosol size dependence of below-cloud scavenging.

Despite all efforts to explore and correct this issue in FLEXPART, there is still a tendency to

underpredict BC measurements in the Arctic. Similarly, in simulations of [137]Cs from the Fukushima accident there is a latitudinal gradient in model bias, with underprediction of observations at high latitudes. For BC, assuming a larger efficiency of the particles to serve as CCN than as IN reduced the Arctic underprediction and also produced a seasonal cycle of BC concentrations that is closer to the observed one, compared to simulations assuming equal CCN and IN efficiency. For [137]Cs, however,

no noticeable improvement in the latitudinal gradient of model bias was found. A reason for this may be that in the large fraction of clouds defined as mixed phase (77%), the Bergeron-Findeisen effect, as represented in Fig. 1, may not be sufficiently strong.

Though there are limitations to the level of sophistication possible for aerosol removal in linear Lagrangian models, the wet removal scheme introduced in FLEXPART is capable of distinguishing

and treating most aspects of wet removal for aerosols with many different characteristics. Our results show that the new scheme produces aerosol lifetimes and concentrations that are realistic when compared with observations.

## 6 Code Availability

FLEXPART is a free software, and can be freely redistributed or modified under the terms of the

GNU General Public License. The FLEXPART v10, coded in Fortran 95 used in this study and also prior versions of FLEXPART are available through http://www.flexpart.eu/ .

*Acknowledgements.* We would like to thank all the scientists who produced the CTBTO measurement data and made them available to us. All absorption data was downloaded through http://ebas.nilu.no/ and we thank the DOE/ARM program and NOAA (J. Ogren), Environment and Climate Change Canada (S. Sharma), Leibniz

Institute for Tropospheric Research (T. Tuch), and Finnish Meteorological Institute (H. Lihavainen) for use of their aerosol light absorption data. This work was supported by NordForsk as part of the Nordic Centres of





Excellence "Cryosphere Atmosphere Interactions in an Arctic Changing Climate" (CRAICC) and "eScience
Tools for Investigating Climate Change at High Northern Latitudes" (eSTICC). C. Groot Zwaaftink was funded
by the Swiss National Science Foundation (P2ELP2_155294). We also thank J. Ogren for valuable discussion
and insights.



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

**Table 1.** Parameters used in Eq. 4 for below cloud scavenging from Laakso et al. (2003) and Kyrö et al. (2009).

| | $a$ | $b$ | $c$ | $d$ | $e$ | $f$ | $C_*$ | $I_o$ | $\lambda_o$ |
|---|---|---|---|---|---|---|---|---|---|
| Laakso | 274.36 | 332839.6 | 226656 | 58005.9 | 6588.38 | 0.24498 | $C_{rain}$ | 1 | 1 |
| Kyrö | 22.7 | 0 | 0 | 1321 | 381 | 0 | $C_{snow}$ | 1 | 1 |





**Table 2.** Sensitivity analysis of lifetimes $\tau_F$ for dust particles of different sizes (diameter d) and with different settings of the removal parameters. (Green: default parameter settings, blue: the longest lifetime if only one deposition process is reduced and red the changed parameter(s) relative to the default settings).

| Parameters | | | | Lifetime (days) | | | | |
|---|---|---|---|---|---|---|---|---|
| $C_{rain}$ | $C_{snow}$ | $CCN_{eff}$ | $IN_{eff}$ | d=0.2 μm | d=2.2 μm | d=6.2 μm | d=10.2 μm | d=18.2 μm |
| 1.00 | 1.00 | 0.15 | 0.02 | 31.8 | 11.6 | 1.8 | 0.8 | 0.25 |
| 1.00 | 1.00 | 0.07 | 0.02 | 39.1 | 12.5 | 1.9 | 0.8 | 0.3 |
| 1.00 | 1.00 | 0.01 | 0.02 | 51.6 | 13.7 | 1.9 | 0.8 | 0.3 |
| 1.00 | 1.00 | 0.15 | 0.01 | 32.7 | 11.6 | 1.8 | 0.8 | 0.3 |
| 1.00 | 1.00 | 0.15 | 0.00 | 33.4 | 11.7 | 1.8 | 0.8 | 0.3 |
| 1.00 | 0.50 | 0.15 | 0.02 | 35.8 | 12.6 | 1.8 | 0.8 | 0.3 |
| 1.00 | 0.10 | 0.15 | 0.02 | 40.5 | 16.3 | 1.8 | 0.8 | 0.3 |
| 0.50 | 1.00 | 0.15 | 0.02 | 34.1 | 12.5 | 1.9 | 0.9 | 0.3 |
| 0.10 | 1.00 | 0.15 | 0.02 | 36.4 | 13.6 | 2.0 | 0.9 | 0.3 |
| 0.10 | 0.10 | 0.01 | 0.00 | 141 | 31.9 | 2.2 | 0.9 | 0.3 |

**Table 3.** Sensitivity analysis of e-folding lifetimes $\tau_e$ for particles of different sizes (diameter d) and with different settings of removal parameters, for the Fukushima case study. The lifetime is also calculated for the total size distribution (Distr., last column). In addition to the lifetime, the relative bias (bias), calculated as the average of all the daily mean concentrations simulated with FLEXPART divided by the observed daily mean concentrations for all days after day 15, is also reported. Cases where the simulated concentrations were too low to reliably estimate lifetime or bias are denoted with LC*.

| # | $C_{rain}$ | $C_{snow}$ | $CCN_{eff}$ | $IN_{eff}$ | d=0.4 μm $\tau_e$ | bias | d=0.65 μm $\tau_e$ | bias | d=1.0 μm $\tau_e$ | bias | d=2.2 μm $\tau_e$ | bias | d=4.0 μm $\tau_e$ | bias | d=6.2 μm $\tau_e$ | bias | Distr. $\tau_e$ | bias |
|---|---|---|---|---|---|---|---|---|---|---|---|---|---|---|---|---|---|---|
| 1 | 1.00 | 1.00 | 0.90 | 0.90 | 11.7 | 18.7 | 10.8 | 11 | 9.6 | 5 | 7.6 | 0.2 | 5.4 | 0.01 | LC* | LC* | 10.08 | 0.99 |
| 2 | 1.00 | 1.00 | 0.40 | 0.40 | 17.9 | 103 | 15.2 | 55.5 | 12.0 | 21.3 | 7.9 | 0.8 | 5.5 | 0.02 | 3.0 | LC* | 13.4 | 4.6 |
| 3 | 1.00 | 1.00 | 0.15 | 0.15 | 25.2 | 192 | 19.2 | 109 | 13.8 | 38 | 8.1 | 1.1 | 4.8 | 0.02 | 2.8 | LC* | 18.6 | 96 |
| 4 | 1.00 | 1.00 | 0.00 | 0.00 | 66.0 | >10³ | | | | | | | | | | | | |
| 5 | 0.00 | 0.00 | 0.90 | 0.90 | | | | | | | 11.0 | 1.3 | | | | | | |



**Table 4.** Aerosol specifications for the eight simulations done for BC. The first four columns report the aerosol removal parameters used, the following columns show the median concentration ($\mathrm{ng\,m^{-3}}$) at each station and the last column reports the median of all modelled values. (blue: the value for each station that is closest to the observed values (bottom row), green: default coefficients, and red: changed parameters)

| | Coefficients | | | | Annual mean Concentration ($\mathrm{ng\,m^{-3}}$) | | | | | | |
|---|---|---|---|---|---|---|---|---|---|---|---|
| # | $C_{rain}$ | $C_{snow}$ | $CCN_{eff}$ | $IN_{eff}$ | MEL | SGP | PAL | BRW | ZEP | ALT | ALL |
| 1 | 1.00 | 1.00 | 0.90 | 0.10 | 700.4 | 234.1 | 33.9 | 7.4 | 9.5 | 6.2 | 33.3 |
| 2 | 1.00 | 1.00 | 0.30 | 0.03 | 736.8 | 252.2 | 61.6 | 10.5 | 16.0 | 8.09 | 58.6 |
| 3* | 1.00 | 1.00 | 0.30 | 0.03 | 713.2 | 245.2 | 45.4 | 8.5 | 9.4 | 7.5 | 45.4 |
| 4 | 1.00 | 1.00 | 9.00 | 1.00 | 428.8 | 113.6 | 4.8 | <0.01 | <0.01 | <0.01 | 1.0 |
| 5 | 10.0 | 1.00 | 0.90 | 0.10 | 615.6 | 194.1 | 20.8 | 3.4 | 4.4 | 3.3 | 22.0 |
| 6 | 1.00 | 0.00 | 0.90 | 0.10 | 690.85 | 232.6 | 36.0 | 8.9 | 10.3 | 6.9 | 40.3 |
| 7 | 1.00 | 1.00 | 0.60 | 0.60 | 673.2 | 219.8 | 30.6 | 6.0 | 8.9 | 6.3 | 31.5 |
| 8 | 1.00 | 1.00 | 0.45 | 0.10 | 727.2 | 244.2 | 37.8 | 8.5 | 10.2 | 7.5 | 41.3 |
| **OBSERVED** | | | | | 366.9 | 211.6 | 36.35 | 17.8 | 11.8 | 19.8 | 19.8 |

* Aerosol diameter was reduced to 20 nm

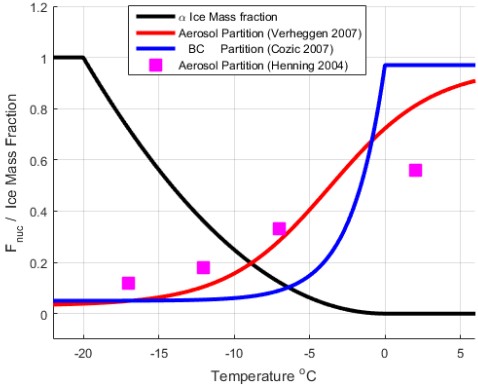

**Figure 1.** The fraction of cloud water that is in the ice phase ($\alpha$) according to equation 1 (black line) and the fraction of aerosols that reside within cloud droplets ($F_{nuc}$) (colored lines and dots) as a function of temperature. For $F_{nuc}$, partitioning values for aerosol number from Verheggen et al. (2007) (red line), from Henning et al. (2004) (magenta dots) and from Cozic et al. (2007) (valid for black carbon (BC) particles) (blue line) are shown. For the BC partitioning, ice fraction was converted to temperature using $\alpha$.





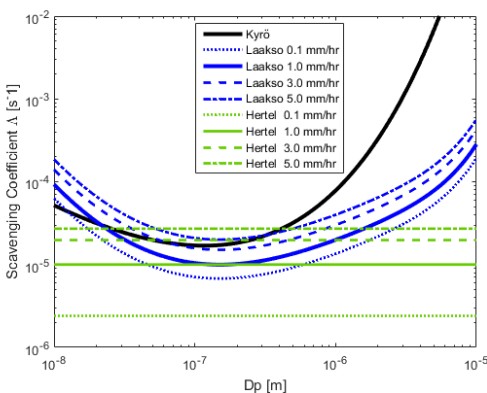

**Figure 2.** Below-cloud scavenging coefficients as a function of aerosol size. Shown are the new parameterizations of Laakso et al. (2003) (blue lines) for rain and Kyrö et al. (2009) (black line) for snow, and the old parameterization of Hertel et al. (1995) used in previous FLEXPART versions with the parameters A=1e-5 and B=0.62 (green). Values are shown for four different precipitation intensities: 0.1 (dotted lines), 1 (solid lines), 3 (dashed lines) and 5 mm/hr (stippled lines).

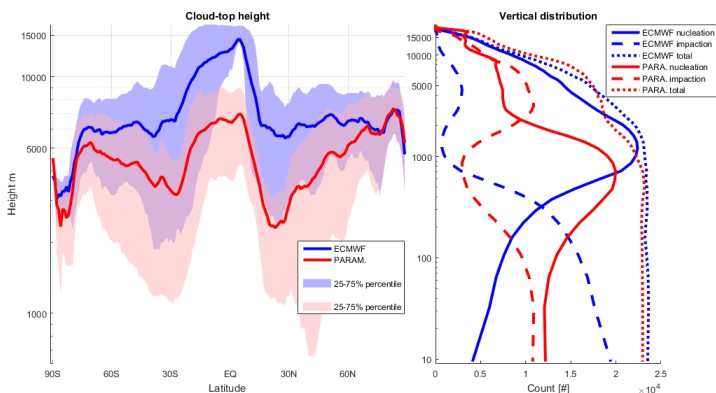

**Figure 3.** Left: The zonally averaged median cloud top heights of precipitating clouds as a function of latitude, averaged over a 90-day period starting in December 2006. Clouds are defined using either the FLEXPART relative humidity-based parameterization (red line) or by CTWC data (blue line). The shaded areas span the 25-75 percentiles. Right: Number of potential removal events globally, for in-cloud nucleation scavenging (solid lines), below-cloud impaction scavenging (dashed lines) and the sum of the two (dotted lines), for both the parameterized clouds (red lines) and when ECMWF cloud water fields are used (blue lines). Note that the height scales are different for the left and right panels.





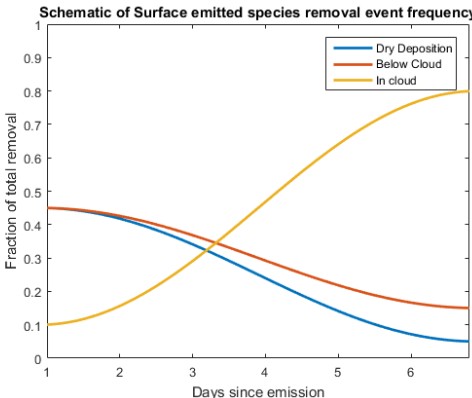

**Figure 4.** Relative frequency of removal events for a pulse of dust emitted in Central Europe on 14 April 2007. For illustration purposes, daily frequencies were fitted with a polynomial.

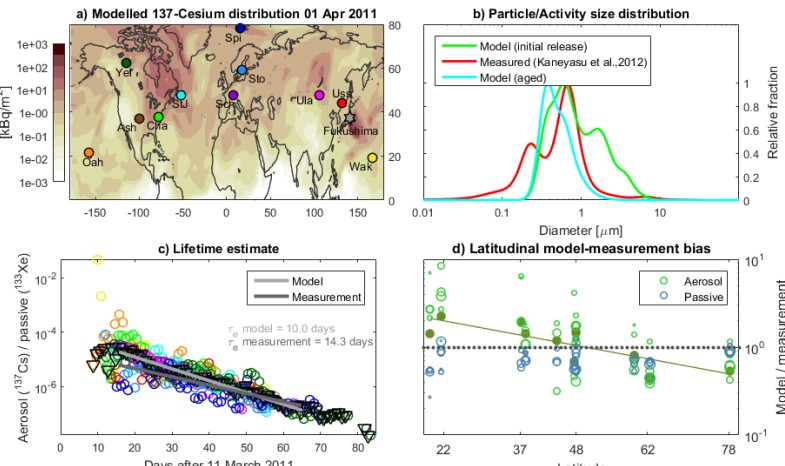

**Figure 5.** a) The concentration of $^{137}$Cs in the Northern Hemisphere on day 15 after the initial release. The locations of the observational sites used in this paper are marked with colored circles. b) Normalized initial (green) and aged (cyan) aerosol surface area distribution of the aerosols used in the simulation. For comparison the measured aerosol size distribution of Kaneyasu et al. (2012) is shown in red. c) Simulated $^{137}$Cs/$^{133}$Xe concentration ratios for the different stations as a function of time after the accident. The circle colors used for the different stations correspond to those used in panel a. The light gray line shows the log-linear fit to the model data. The dark gray line shows the fit to the observed concentrations (see Kristiansen et al., 2015). d) Ratio of modeled to observed concentrations as a function of latitude for the passive tracer $^{133}$Xe (blue circles) and the aerosol-bound $^{137}$Cs (green circles).





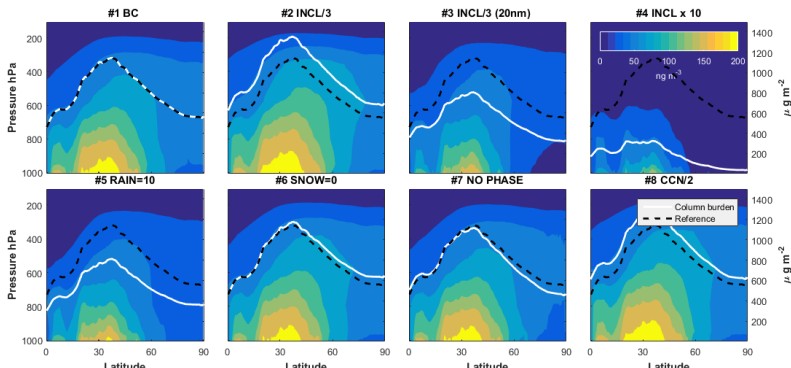

**Figure 6.** Northern Hemisphere vertical distribution of BC for eight different settings of the removal parameters. Top left panel shows the concentrations for the reference settings for BC. The other panels show results of the sensitivity simulations (see Table 4 for details). Five vertical layers were used and the horizontal resolution is 0.5 degrees. The white line is the latitudinal column burden for each simulation and the dashed black line repeats the results for the reference BC simulation, for comparison purposes.

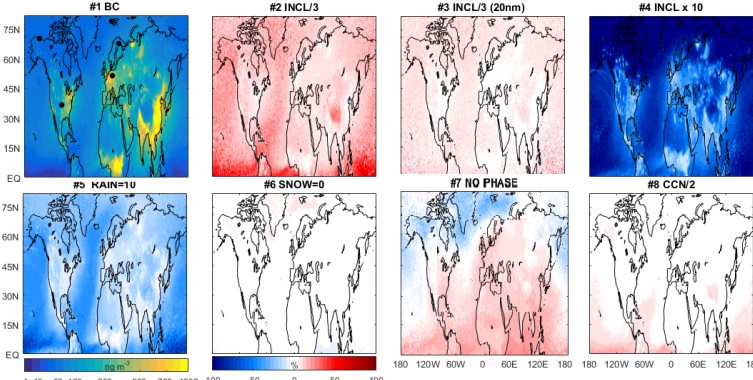

**Figure 7.** Annual average BC concentration in the lowest model layer (0-100 m) for the reference simulation (top left) for the year 2007. Black circles mark the locations of the measurement stations used for model comparisons. The other panels show the relative difference to this reference version (in %) for the seven other simulations using parameter settings from Table 4.





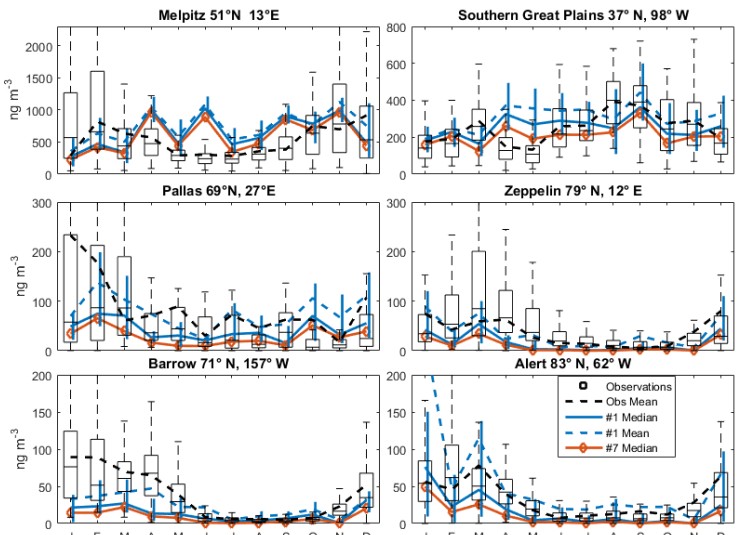

**Figure 8.** Modeled and observed monthly BC concentrations at six different measurement stations for the reference BC simulation. The black boxes cover the 25-75% percentile range, the black horizontal line the median, and the black whiskers the 10-90% percentile range of the observations. Modeled median values are plotted in blue with vertical lines showing the 25-75% percentile range. The stippled blue line shows the model mean. Also shown are the median values obtained from simulation #7.





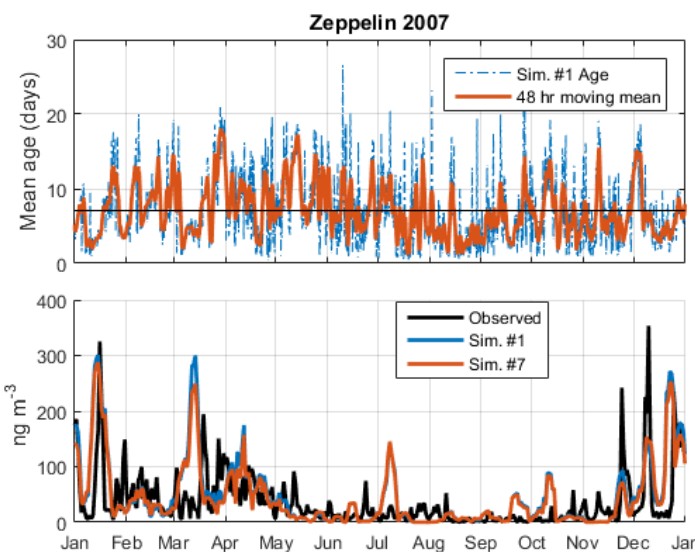

**Figure 9.** Age of FLEXPART BC aerosol for simulation #1 at Zeppelin every 6 hours (blue) and smoothed with a 48-hour running mean (red) for the year 2007 (top panel). The black line shows the annual mean age of 7.7 days. The bottom panel shows the simulated (blue) and observed (black) BC concentrations, smoothed with a 48-hour running mean, for 2007.