# Peer review of "A new aerosol wet removal scheme for the Lagrangian particle model FLEXPART v10"

_Geoscientific Model Development, 2016_

## Short Comment (SC1) · 17 Nov 2016

Dear authors,

In my role as Executive editor of GMD, I would like to bring to your attention our Editorial version 1.1:

http://www.geosci-model-dev.net/8/3487/2015/gmd-8-3487-2015.html

This highlights some requirements of papers published in GMD, which is also available on the GMD website in the 'Manuscript Types' section:

http://www.geoscientific-model-development.net/submission/manuscript_types.html

In particular, please note that for your paper, the following requirement has not been met in the Discussions paper:

[Figure]

- "The main paper must give the model name and version number (or other unique identifier) in the title."

Please add the version number of FLEXPART in the title upon your revised submission to GMD.

Yours,

Astrid Kerkweg

---

## Referee Comment (RC1) · Anonymous Referee #1 · 29 Nov 2016

General comments: This paper describes the incorporation of a new aerosol wet scavenging scheme into the FLEXPART model. The authors included a new parameterization for wet removal within and below clouds considering the physical state of water in the clouds and the precipitation type. This parameterization was set for three different types of aerosols and compared against available measurements. A series of sensitivity analysis were also performed to test the range of results obtained under different parameterization assumptions.

Specific comments:

- Although it is very commendable the inclusion of comparisons between measurements and model results, I have serious concerns regarding the uncertainties in other processes (e.g. emissions) that might hinder the conclusions reached by this work.

Consequently, to give the readers a sense of the relative changes introduced by the new parameterizations, I would suggest including the results from the old parameterization for each of the three applications presented in this work.

- Line 160. How did you come up with a value of 6.1 for icr? Is this basically an empirical factor?

- Including a list of recommended values for the parameterizations for different aerosols will enhance the value of this work.

Technical corrections:

Line 40. Please add chemical processes for completeness in the sentence.

Line 85-86. HYSPLIT has a new option for in-cloud wet scavenging parameterization (See Stein et al 2015, supplement). NAME has also updated its wet deposition scheme (see http://www.metoffice.gov.uk/media/pdf/c/a/FRTR584.pdf)

Line 91 non-linear chemistry has been included in this kind of models (e.g. Chock, D. P., and S. L. Winkler, 1994: A particle grid air quality modeling approach: 1. The dispersion aspect. J. Geophys. Res., 99, 1019–1031, doi:10.1029/93JD02795. Chock, D. P., and S. L. Winkler, 1994b: A particle grid air quality modeling approach: 2. Coupling with chemistry. J. Geophys. Res., 99 (D1), 1033–1041, doi:10.1029/93JD02796.)

Line 247. Sulfate is not a primary aerosol. Please correct the sentence.

Line 520- 524 This is very speculative. There is no empirical evidence that this is why the model shows a latitudinal bias.

---

## Referee Comment (RC2) · Anonymous Referee #2 · 10 Jan 2017

**1   general comments**

The authors discuss a new parametrization for FLEXPART of both in-cloud and below-cloud wet scavenging of aerosols. The parametrization has revised the way that cloud information is obtained and processed and considers both size dependent, aerosol dependent and phase dependent scavenging. Results from the scheme are compared against observations and results from other published studies for three different aerosols. The sensitivity of the parametrization to the various parameters of the scheme is further assessed.

I found this an interesting paper which is well written and comprehensive.

**2 specific comments**

1. There is a good, explicit and easy to understand summary of what the new aerosol wet removal scheme includes but I'd like the authors to emphasise further what improvements this new scheme gives over the previous scheme (i.e., which of the features discussed are new), particularly since some discussion of results with the old and new schemes are included.

2. I would disagree with the authors' comments regarding aerosol schemes in Lagrangian models (lines 83-86). Whilst these schemes may have remained constant for some time, developments and reviews have taken place since their first introduction. For example, NAME has recently had a new size dependent wet deposition scheme added for particles.

3. Presumably some critical value of $CTWC$ is used to determine cloud ($CTWC$ > some value). What is this value and how much certainty is there in it? How sensitive are results to this value?

4. Scavenging from multiple layers of cloud may be difficult to accurately model with only surface precipitation data. For example, the precipitation rates / intensities are likely to be different from clouds at different layers. Furthermore, what happens between layers? Is this considered as "below cloud" scavenging given that all layers of cloud are considered to be precipitating?

5. Where does the value of 6.1 for $ic_r$ come from (line 160)? Is it tuned to data? It sounds like results are quite sensitive to the value of $ic_r$ (lines 162-163).

6. The authors might like to mention the Greenfield gap (i.e., that there is a range of particle sizes which are neither efficiently collected by Brownian motion nor impaction) (lines 202-203).

7. The authors use either the surface precipitation type or a surface temperature to determine whether the Laakso or Kyro parametrization should be used. However, precipitation which is rain at the surface may be snow at the point of impaction (or vice versa). Hence it would seem more appropriate to use the temperature at the aerosol height. Also, how is the water phase of the clouds determined (lines 357-361)? Is this determined from the temperature and/or precipitation type? And if so, can the authors comment on the height of data that is used and the appropriateness of this data if surface data is used to determine the phase of elevated clouds?

8. Where do the values for $CCN_{eff}$ and $IN_{eff}$ come from? Can the authors add some references in addition to those given for black carbon? For soluble aerosols Croft et al. (Atmos. Chem. Phys., 10, 1511–1543, 2010) have a large difference between values for liquid and ice stratiform clouds (albeit similar values for convective) which contrasts here to the default values in Table 3 for $^{137}Cs$ attached to sulfate. I refer the authors to their comment on lines 517-518 questioning whether their in-cloud scavenging in ice clouds is too effective for this aerosol. And lastly, how is it possible for $CCN_{eff}$ (a fraction) to take values > 1 (Table 4)?

**3  Technical corrections**

1. Line 121 – There is no paper by Stohl et al. from 2016 listed in the references. Is the date here incorrect?

2. An explanation of $C^*$ should be given when equation 4 is introduced.

3. BRW should be BAR (line 321) and Table 4 (or maybe BAR should be BRW on lines 315 and 321).

4. Could the authors label day 7 on Figure 4.

5. A couple of comments regarding Table 2: (i) For 18.2 $\mu$m the lifetime for the default parameter settings is given to 2 decimal places, whereas for all other parameter settings, it is only given to 1 decimal place. Furthermore, if the lifetime for the default parameter settings was given to 1 decimal place, it would be the same (0.3) as the lifetimes given for other parameter settings so it is not obvious that there is any change. Is there? Perhaps there should be a consistent use of decimal places here. (ii) The longest lifetime if only one deposition process is reduced is indicated in blue but yet there are many such cases with the same lifetimes. Should these also be indicated in blue?

6. I think the reference to Table 1 on line 421 should be to Table 2.

7. There is an inconsistency in the lifetime increases on line 424. The 20% increase (6.2 $\mu$m particles) means that the end result is 120%. The comparison is for a 350% increase (not a 450% increase as stated – this is the end result).

8. The footnote on page 13 states that $ic_r = 6.2$ is used in this paper, whereas the value stated on line 160 is 6.1

9. Is there a typo in the figure caption of Figure 5b? Should "surface area distribution" be "size distribution"?

10. There is a typo in line 473. 10.8 days is for the 0.65 $\mu$m bin size (or alternatively 11.7 days for the 0.4 $\mu$m bin size).

11. Table 4: The column header refers to "mean" concentration, whereas the caption refers to the "median" concentration.

12. Line 550: "concentration" should be "column burden".

13. Line 574: Reference to Fig 6 should be Fig 7.

[Figure]

**4 Minor issues**

Some further minor issues and questions which the authors may also like to address.

1. Lines 4-6 implies that differentiating between cloud water phases allows an aerosol type dependent removal scheme. These seem two independent things.

2. It is not clear whether the reference to "dry deposition" in most places includes "gravitational settling". In the abstract (lines 19-20) the two are referred to separately (Dry deposition and gravitational settling) but gravitational settling isn't always referred to elsewhere and hence one is unclear whether references to dry deposition intend to include gravitational settling.

3. It would help to briefly define "aerosols" as this term is sometimes misused and confused with the generic term "particles".

4. What is the meaning of "outside a cloud" (line 328)? In particular, are not 'below' or 'above' clouds 'outside' of clouds?

5. It seems a little counter-intuitive that there could be more below-cloud impaction scavenging events around 5000 m using the FLEXPART relative humidity parametrization (Figure 3b) when the cloud-top height is, in general, lower using this scheme. The authors mention multiple cloud layers being the reason for the small peak in the below cloud impaction events at this height using the ECMWF parametrization, and that in reality there is probably not scavenging at this level due to the upper level clouds being non-precipitating. Do the same reasons apply to the peak in below-cloud scavenging events at 5000 m seen when using the FLEXPART relative humidity parametrization and why does this scheme give a larger count? The smaller count for the ECMWF parametrization, if the below cloud scavenging events at 5000 m are not real, may be indicative of better performance of the ECMWF parametrization.

[Figure]

6. Is mention of "particles in the 0.2 – 0.6 $\mu$m" (line 404) intended to be a general reference to accumulation mode particles or a reference to particle sizes modelled here? If it is the latter, I am having trouble matching this with the numbers in Table 2 which gives a diameter of 0.2 $\mu$m but the next size up is 2.2 $\mu$m.

7. Nanometre and micrometre are used interchangeably (e.g. 0.2 $\mu$m and 200 nm are both used). Some consistency referring to particles of the same size would be better.

8. In the comparison of the measured and model size distribution (lines 447-448), the authors say that the measured peak at 1 $\mu$m matched the model well. The aged model peak is at a little lower particle size distribution.

9. What are the upside down black triangles in Figure 5c? Are these the "daily median ratios" (line 455)?

10. The mention of "aerosol type" on line 512 refers, I think, to splitting the aerosol size distribution up into bins with different particle sizes and scavenging rates. If this is correct, I find this terminology confusing since I would say that it is one aerosol type (i.e. $^{137}Cs$ attached to sulphate) and not more than one aerosol type.

11. Is the dotted black line in Figure 5d the 1-1 line or the line of best fit to the $^{133}Xe$ (or both?)?

12. I don't follow the final sentence of the paragraph on lines 559-561. I cannot see a burden with a substantially different dependence on latitude in Figure 6. Furthermore, simulations 5 and 8 also have a phase dependent change to the removal parameters.

13. Lines 570-571. The authors refer to an increase in remote areas like the Arctic but the increase seems much larger near the equator.

14. It is difficult to see the black circles in Figure 7, particular since the coastlines are also black.

---

## Author Comment (AC1) · 18 Feb 2017

**SC #1 A. Kerkweg**

Please add the version number of FLEXPART in the title upon your revised submission to GMD.

Version number added to title, new title:
*A new aerosol wet removal scheme for the Lagrangian particle model FLEXPART v10*

---

## Author Comment (AC2) · 18 Feb 2017

We would like to thank both reviewers for their detailed and constructive comments on our manuscript. It is very much appreciated that both reviewers took such obvious care and gave excellent comments. It is our hope that they agree that the changes introduced in the new version based on their comments and suggestions, have helped improve the work. Where comments from both reviewers address the same issue, one answer is given for both comments. Below, we list the reviewer comments and our corresponding replies (in blue) as well as excerpts from the new draft (blue italic). A supplement of tracked changes is also provided.

**RC #1**

General comments: This paper describes the incorporation of a new aerosol wet scavenging scheme into the FLEXPART model. The authors included a new parameterization for wet removal within and below clouds considering the physical state of water in the clouds and the precipitation type. This parameterization was set for three different types of aerosols and compared against available measurements. A series of sensitivity analysis were also performed to test the range of results obtained under different parameterization assumptions.

Specific comments:

- Although it is very commendable the inclusion of comparisons between measurements and model results, I have serious concerns regarding the uncertainties in other processes (e.g. emissions) that might hinder the conclusions reached by this work. Consequently, to give the readers a sense of the relative changes introduced by the new parameterizations, I would suggest including the results from the old parameterization for each of the three applications presented in this work.

This is a good and valid concern. Therefore, as suggested we have added some results from version 9 of FLEXPART to each of the 3 sensitivity studies. The following changes were done to accommodate this:

1. In Figure 5c: The ratio of 137CS/133Xe from Kristiansen et al., (2015) using FLEXPART version 9 has been added to show the difference in e-folding timescale from previous versions. In Figure 5d: a fit to the latitudinal bias of FLEXPART v9 is now also shown.

2. For mineral dust, a line was added to Table 2 that reports $\tau_F$ using the standard
removal in FLEXPART v9.

3. In Figure 9, we have replaced simulation #7 concentrations obtained with version 9 of FLEXPART. The removal used is however, the removal used for BC in previous publications, which is a modified version of the existing scheme in FLEXPART.

When adding these results, also some small changes to text were made throughout the document to incorporate the results in the text (see supplement).

- Line 160. How did you come up with a value of 6.1 for icr? Is this basically an empirical factor?

The empirical nature of the value of $i_{cr}$ used was perhaps not expressed explicitly enough. Though $i_{cr}$ should be representative of that there is cloud water replenishment, linking the time averaged cloud water fields to precipitation rates, the value of 6.1 is a purely empirical factor in FLEXPART. Values suggested in literature for cloud water replenishment suggest 15-120 min for warm marine stratiform drizzle (Wood et al., 2009). The value of replenishment is closely tied to cloud droplet autoconversion rates (Khairoutdinov and Kogan 2000) which are not as well constrained for mixed and ice phase clouds and not resolved in FLEXPART.

In text it was clarified that $i_{cr}$ indeed is an empirical value:

*where $F_{nuc}$, the nucleation efficiency, is the fraction of the aerosol within the cloud that is in the cloud water (see Fig. 1). While $ic_r$ represents the cloud water replenishment rate, it cannot be determined from the ECMWF output data. Therefore, the determination of the constant $ic_r$ was done on the basis of empirical testing in FLEXPART and must be considered a tuning parameter.*

*Compared to the previous FLEXPART scheme described in Stohl et al., 2005,*

*$ic_r/PCW$ replaces the cloud water representation that was calculated based on an empirical relationship with precipitation rate ($cl = 2 \ 10^{-7} \ I^{0.36}$). The overall best results were obtained for $ic_r$ set to a value of 6.1 for the ECMWF cloud water fields, which is used for all simulations in this paper. This resulted in a somewhat slower in-cloud removal rate with the new compared to the old parametrisation. Comparison of the two parametrisations also shows that using $ic_r/PCW$ gives overall weaker dependence on $I$, compared to $cl$ in the old removal scheme. For simulations where in-cloud removal constitutes a large fraction of the removal, i.e. especially for soluble accumulation mode aerosols, the empirical value of $ic_r$ has a large impact on overall removal rates.*

- Including a list of recommended values for the parameterizations for different aerosols will enhance the value of this work.

This is a good suggestion which we wholeheartedly agree to. A paragraph has been added in "Conclusions" to this effect:

*Simulations for the accumulation mode particles with FLEXPART are highly sensitive to the choice of $CCN_{eff}$ and $IN_{eff}$ values, which describe the particles' efficiency to serve as cloud condensation and ice nuclei. Overall, it was found that the sum of $CCN_{eff} + IN_{eff}$ is more important for the removal efficiency than the individual choice of values for $CCN_{eff}$ or $IN_{eff}$. For the three aerosol types, we recommend the following values: Regarding insoluble aerosols, Zwaaftink et al., 2016 found good agreement between modeled and observed concentrations when using $CCN_{eff} = 0.15$ and $IN_{eff} = 0.02$ for mineral dust. For BC, $CCN_{eff} = 0.9$ and $IN_{eff} = 0.1$ gave the overall best results, and these values are also comparable with what was found by Cozic et al., 2007. Soluble aerosol ($^{137}Cs$) concentrations compared best with $CCN_{eff} = 0.9$ and $IN_{eff} = 0.9$. The latter value is somewhat higher than $IN_{eff}$ values suggested by measurements of e.g. Henning et al., 2004.*

Technical corrections:

Line 40. Please add chemical processes for completeness in the sentence.

Thank you, this is now changed.

Line 85-86. HYSPLIT has a new option for in-cloud wet scavenging parameterization (See Stein et al 2015, supplement). NAME has also updated its wet deposition scheme (see http://www.metoffice.gov.uk/media/pdf/c/a/FRTR584.pdf)

The paragraph has been updated so that it now more correctly reflects that the wet removal of NAME and HYSPLIT has been updated. We now write:

*The aerosol removal scheme in FLEXPART Hertel et al., 1995 has remained relatively unchanged since its incorporation in the late 1990s. Other, similar Lagrangian models like NAME and HYSPLIT have had recent updates to their aerosol removal (Webster et al., 2014; Stein et al., 2015). However, the overall level of detail also in these models remains low compared to known theory (e.g. Feng 2007). One reason for this is the limiting factors that constrain the possible ways of treating aerosol removal within the Lagrangian model framework.*

Line 91 non-linear chemistry has been included in this kind of models (e.g. Chock, D. P., and S. L. Winkler, 1994: A particle grid air quality modeling approach: 1. The dispersion aspect. J. Geophys. Res ., 99, 1019–1031, doi:10.1029/93JD02795. Chock, D. P., and S. L. Winkler, 1994b: A particle grid air quality modeling approach: 2. Coupling with chemistry. J. Geophys. Res., 99 (D1), 1033–1041, doi:10.1029/93JD02796.)

We have rephrased the paragraph to show that more extensions to the linear model concept exist. We now write:

[Figure]

*A main consideration within this framework is that each transported computational particle is independent of others. Extensions of this concept to allow for non-linear chemistry exist (Chock et al., 1994a,b), also for FLEXPART (Cassiani et al., 2013), but the reference version of FLEXPART is a purely linear transport model. Within such a linear model, it is impossible to include aerosol processes which depend on the aerosol concentration (e.g., coagulation or non-linear chemical reactions). Furthermore, to facilitate consistency between forward and backward runs of FLEXPART, parameterizations that depend on the age of the aerosol (i.e. time after emission for primary aerosols) should be avoided as well. This limits the level of sophistication that can be incorporated into an aerosol removal scheme. Nevertheless, a realistic treatment of aerosols is possible even with these limitations.*

Line 247. Sulfate is not a primary aerosol. Please correct the sentence.

Yes that is correct, thank you, this is now changed so it no longer reads that sulphate is a primary aerosol.

Line 520- 524 This is very speculative. There is no empirical evidence that this is why the model shows a latitudinal bias.

Agreed. Results presented in the paper do not show any evidence for this, and it is thus a speculation. The paragraph has changed to reflect this. "The probable cause" is changed to "One of the possible causes". Also added a further sentence on other possible causes of the latitudinal dependence of model/measurement bias to highlight that the reason for the latitudinal bias. We have now written:

*In Fig. 5d the mean model / observed concentration ratios at the different stations are plotted against latitude. A prominent feature of FLEXPART and indeed most models used by Kristiansen et al., 2015 is a tendency to overpredict concentrations at low latitudes and underpredict concentrations at high latitudes. This tendency is also present*

*with the new removal scheme, where model / observation ratios decrease with latitude. The green line shows a logarithmic fit to the station median data. The same fit was done to the mean from a simulation using FLEXPART version 9 (pink). This shows that the new model, while still having a systematic latitudinal dependence, represents a clear improvement over the old version. One possible explanation of the decreasing model/observation ratios with latitude might be that in-cloud scavenging in ice clouds is too effective. However, sensitivity simulations where only $IN_{eff}$ was reduced (not shown) revealed that this change had only a small effect in further reducing the latitudinal bias. One of the possible causes of this is the high proportion of mixed phase clouds (77%) which reduces the impact of the latitudinal dependence of the frequency of ice-phase clouds after that much time for an emission pulse. Another possibility is that cloud phase is not well captured by the ECMWF model, as in many other models Cesana et al., 2015. It may also be relevant that the clouds have on average higher cloud tops near the equator, so that temperature and thus the mixing state of clouds does not have a strong enough latitudinal dependence in the Northern Hemisphere at the time of this simulation (March-May).*

---

## Author Comment (AC3) · 18 Feb 2017

We would like to thank both reviewers for their detailed and constructive comments on our manuscript. It is very much appreciated that both reviewers took such obvious care and gave excellent comments. It is our hope that they agree that the changes introduced in the new version based on their comments and suggestions, have helped improve the work. Where comments from both reviewers address the same issue, one answer is given for both comments. Below, we list the reviewer comments and our corresponding replies (in blue) as well as excerpts from the new draft (blue italic). A supplement of tracked changes is also provided.

**RC #2**

1 general comments

The authors discuss a new parametrization for FLEXPART of both in-cloud and below-cloud wet scavenging of aerosols. The parametrization has revised the way that cloud information is obtained and processed and considers both size dependent, aerosol dependent and phase dependent scavenging. Results from the scheme are compared against observations and results from other published studies for three different aerosols. The sensitivity of the parametrization to the various parameters of the scheme is further assessed.

I found this an interesting paper which is well written and comprehensive.

2 specific comments

1. There is a good, explicit and easy to understand summary of what the new aerosol wet removal scheme includes but I'd like the authors to emphasise further what improvements this new scheme gives over the previous scheme (i.e., which of the features discussed are new), particularly since some discussion of results with the old and new schemes are included.

It is apparent that some further discussion and comparison with the old removal scheme is needed. This is in line with RC#1 and warrants minor changes several places throughout the paper, including adding some results from FLEXPART version 9. We agree that this improves the clarity of what changes have actually been done.

In section 2.1 "Clouds and precipitation in FLEXPART" it is now clearly stated that the reading of clouds from the prescribed meteorology is a new feature. The advantage of

using these ECMWF clouds over the old, RH based scheme are discussed in section
"4.1 Wet scavenging event statistics"

The new features of the in-cloud removal scheme are now presented as new features,
and it is inserted what they are in section 2.3. The new details added to in-cloud
removal are related to the components that make up $F_{nuc}$ ($IN_{eff}$, $CCN_{eff}$ and $\alpha$).
It is also stated that $i_{cr}$ and PCW replaces the old parameterization for cloud water
content

It is also clearly stated now that the new below cloud scheme is in fact introduced in
FLEXPART model version 10 and is a new feature.

In terms of comparison of the two model versions with observations we have added
text and results (see our reply to RC#1 )

2. I would disagree with the authors' comments regarding aerosol schemes in La-
grangian models (lines 83-86). Whilst these schemes may have remained constant for
some time, developments and reviews have taken place since their first introduction.
For example, NAME has recently had a new size dependent wet deposition scheme
added for particles.

The paragraph has been updated so that it now more correctly reflects that the wet
removal of NAME and HYSPLIT has been updated (see our reply to RC#1 ).

3. Presumably some critical value of CTWC is used to determine cloud (CTWC  some
value). What is this value and how much certainty is there in it? How sensitive are
results to this value?

see our response below (RC#2 specific comment 4.)

4. Scavenging from multiple layers of cloud may be difficult to accurately model with

only surface precipitation data. For example, the precipitation rates / intensities are likely to be different from clouds at different layers. Furthermore, what happens between layers? Is this considered as "below cloud" scavenging given that all layers of cloud are considered to be precipitating?

3 & 4. Indeed, it is difficult to accurately model these processes with only surface precipitation data. 3-d precipitation fields would be needed to better capture this, but those are not available in FLEXPART. Therefore, we had to implement a relatively simple scheme. The "some critical value" is $>0$. The main reason for this is that any value chosen here would be incorrect. The value chosen would have to be dependent on factors such as; cloud extent in the grid, whether you are close to the top or bottom of the cloud, type of precipitation, temperature. Some testing was done initially, where the main aim to remove clouds with multiple layers. Though partly successful in this, having the critical value $>0$ had other unintended consequences such as precipitation without clouds which was deemed harder to correct for than actual problems imposed by having it set to 0. If there are two inconsecutive layers of cloud there would be below cloud removal in between the cloud layers. We have clarified the criteria for determining in-cloud or below-cloud scavenging by adding the sentence:

*If $PCW > 0$ in-cloud scavenging is applied.*

5. Where does the value of 6.1 for icr come from (line 160)? Is it tuned to data? It sounds like results are quite sensitive to the value of icr (lines 162-163).

For the empirical nature of $i_{cr}$, see our answer to #RC1 comment 2
As you correctly point out, results for aerosols with a large fraction of in-cloud removal are quite sensitive to the set value of $i_{cr}$. However, it is a linear parameter affecting all in-cloud removal the same. The water washout ratio i.e. the fraction of a clouds' water that is precipitating at a given time will vary between individual clouds. As is discussed already in text, if only in-cloud removal rate is efficient, lifetime, and thus atmospheric
concentrations is very sensitive to in-cloud removal rate. For aerosols where also other removal mechanisms are efficient this sensitivity is much lower.

6. The authors might like to mention the Greenfield gap (i.e., that there is a range of particle sizes which are neither efficiently collected by Brownian motion nor impaction) (lines 202-203).

Yes this is now explicitly mentioned, and not just referred to by citation.

7. The authors use either the surface precipitation type or a surface temperature to determine whether the Laakso or Kyro parametrization should be used. However, precipitation which is rain at the surface may be snow at the point of impaction (or vice versa). Hence it would seem more appropriate to use the temperature at the aerosol height. Also, how is the water phase of the clouds determined (lines 357-361)? Is this determined from the temperature and/or precipitation type? And if so, can the authors comment on the height of data that is used and the appropriateness of this data if surface data is used to determine the phase of elevated clouds?

There was a misunderstanding. Indeed, local temperature at the aerosol altitude is used to calculate both the cloud phase and the precipitation phase of water. If CTWC (=CLWC + CIWC) is used the cloud ice – water partition is temperature dependent, and shown in Fig. 1 as $\alpha$. If CLWC + CIWC fields are used, the $\alpha$ in eq (3) is instead taken directly from these fields. We have now written that we:

*In this study we have used a local temperature threshold of $0^{\circ}C$ is to distinguish between rain and snow, but it is also possible to use rain and snow precipitation intensity read directly into the model from ECMWF analysis data.*

8. Where do the values for CCNeff and INeff come from? Can the authors add some references in addition to those given for black carbon? For soluble aerosols Croft et al.

(Atmos. Chem. Phys., 10, 1511–1543, 2010) have a large difference between values for liquid and ice stratiform clouds (albeit similar values for convective) which contrasts here to the default values in Table 3 for 137Cs attached to sulfate. I refer the authors to their comment on lines 517-518 questioning whether their in-cloud scavenging in ice clouds is too effective for this aerosol. And lastly, how is it possible for CCNef f (a fraction) to take values > 1 (Table 4)?

Indeed, the reviewer is right that INeff and CCNeff are fractions and should strictly have values between 0 and 1. All our recommended values are also in this range. However, for the purpose of a sensitivity study we artificially increased CCNeff to a value of 9. We do not claim that this is a realistic value that should be used in FLEXPART simulations. This was done only to explore what would happen if there was a mechanism that makes aerosols extremely effective as CCN (e.g., a mechanism that increases the number of aerosols).

The high $IN_{eff}$(=0.9) applied for soluble aerosols are indeed higher than what was used in ECHAM5. Building mostly on the same Jungfraujoch measurements the in-cloud removal by (Hoose et al., 2008) used in this ECHAM version has a much lower scavenging efficiency for ice than water for soluble aerosols. However, for our comparison with measurements ($^{137}$Cs), this resulted in a significant overestimation of observed concentrations.

3 Technical corrections

1. Line 121 – There is no paper by Stohl et al. from 2016 listed in the references. Is the date here incorrect?

It is a technical description of FLEXPART v10 in preparation, and the year was missing from the reference. The paper will be submitted to GMDd soon. While not submitted,

we did not want to repeat all technical aspects given in that paper (e.g., on reading input data), also here, and thus refer to it despite the current status.

2. An explanation of $C_*$ should be given when equation 4 is introduced.

The scalar $C_*$ is now introduced with the equation.

3. BRW should be BAR (line 321) and Table 4 (or maybe BAR should be BRW on lines 315 and 321).

Thank you. We will stick to the station acronyms used in GAW (global atmospheric watch), BRW should now be applied everywhere.

4. Could the authors label day 7 on Figure 4.

x-axis label inserted.

5. A couple of comments regarding Table 2: (i) For 18.2 $\mu m$" the lifetime for the default parameter settings is given to 2 decimal places, whereas for all other parameter settings, it is only given to 1 decimal place. Furthermore, if the lifetime for the default parameter settings was given to 1 decimal place, it would be the same (0.3) as the lifetimes given for other parameter settings so it is not obvious that there is any change. Is there? Perhaps there should be a consistent use of decimal places here. (ii) The longest lifetime if only one deposition process is reduced is indicated in blue but yet there are many such cases with the same lifetimes. Should these also be indicated in blue?

(i) The number of decimal places has been changed so that all numbers are consistently given to the accuracy of 1 decimal place.
(ii) We have however left the colouring even though the difference in lifetime is smaller

(sometimes very small indeed) than the accuracy of the listed value in the paper. We feel that this is an acceptable illustration as it also indicate that removal by warm phase precipitation has growing importance with growing aerosol size, a new feature of the removal scheme.

6. I think the reference to Table 1 on line 421 should be to Table 2.

Yes, you are correct, thank you, this is now corrected.

7. There is an inconsistency in the lifetime increases on line 424. The 20% increase (6.2 $\mu$m particles) means that the end result is 120%. The comparison is for a 350% increase (not a 450% increase as stated – this is the end result).

Thank you, this inconsistency have now been corrected.

8. The footnote on page 13 states that icr = 6.2 is used in this paper, whereas the value stated on line 160 is 6.1

Icr = 6.1 is correct this has been changed.

9. Is there a typo in the figure caption of Figure 5b? Should "surface area distribution" be "size distribution"?

Yes, but surface area distribution is correct. To compare the surface activity distribution, surface area distribution was calculated from the aerosol particle mass distribution. Figure caption has been updated.

10. There is a typo in line 473. 10.8 days is for the 0.65 $\mu m$" bin size (or alternatively 11.7 days for the 0.4 $\mu m$" bin size).

Yes, that was a typo, this is now corrected.

11. Table 4: The column header refers to "mean" concentration, whereas the caption refers to the "median" concentration.

The caption was correct the column headers are now corrected, so it all displays "median". Thank you.

12. Line 550: "concentration" should be "column burden".

Yes, thank you, this is now corrected.

13. Line 574: Reference to Fig 6 should be Fig 7.

Yes, thank you, this is now corrected.

4 Minor issues

Some further minor issues and questions which the authors may also like to address.

1. Lines 4-6 implies that differentiating between cloud water phases allows an aerosol type dependent removal scheme. These seem two independent things.

Sentence has been clarified, so this should now be clear that these are indeed separate. We now write:

*The new in-cloud nucleation scavenging depends on cloud water phase (liquid, ice or mixed-phase), based on the aerosol's prescribed efficiency to serve as ice crystal nuclei and liquid water nuclei, respectively.*

2. It is not clear whether the reference to "dry deposition" in most places includes "gravitational settling". In the abstract (lines 19-20) the two are referred to separately

(Dry deposition and gravitational settling) but gravitational settling isn't always referred to elsewhere and hence one is unclear whether references to dry deposition intend to include gravitational settling.

Throughout the text now dry deposition and gravitational settling are used independently, and "dry removal" is used for both collectively.

3. It would help to briefly define "aerosols" as this term is sometimes misused and confused with the generic term "particles".

Throughout text, the term "Aerosol particle" now replaces "particle" when in fact an aerosol particle is meant. In general, "particle" now only refers to FLEXPART particles. We hope this improve the readability of the documnet.

4. What is the meaning of "outside a cloud" (line 328)? In particular, are not 'below' or 'above' clouds 'outside' of clouds?

Replaced *"Outside a cloud"* with *"cloud free column"*

5. It seems a little counter-intuitive that there could be more below-cloud impaction scavenging events around 5000 m using the FLEXPART relative humidity parametrization (Figure 3b) when the cloud-top height is, in general, lower using this scheme. The authors mention multiple cloud layers being the reason for the small peak in the below cloud impaction events at this height using the ECMWF parametrization, and that in reality there is probably not scavenging at this level due to the upper level clouds being non-precipitating. Do the same reasons apply to the peak in below-cloud scavenging events at 5000 m seen when using the FLEXPART relative humidity parametrization and why does this scheme give a larger count? The smaller count for the ECMWF parametrization, if the below cloud scavenging events at 5000 m are not real, may be indicative of better performance of the ECMWF parametrization.

When there is surface precipitation, scavenging is applied in FLEXPART from the top-most grid box that contains cloud water (or RH $> 80\%$) to the surface. Therefore the number of total removal events will be decreasing with altitude. However the density correction of each layer done, may slightly alter this count in Fig 3(right). Below the cloud top, the partitioning between in-cloud and below-cloud is thus solely dependent on whether a cloud is defined inside each individual grid box. The decreased number of both "surface clouds" and high altitude "below-cloud removal" does indeed, in our opinion, indicate a better performance of the ECMWF scheme.

In this context we also found it worth noting that from 5 to 15km altitude the vertical extent of the grid boxes increase significantly and so the count of the highest altitude clouds of the RH based scheme falls outside the 25-75 percentile of the cloud tops (in Figure 3 (left)) as they are distributed on most latitudes and not concentrated in the tropics like in the clouds of the ECMWF defined clouds.

6. Is mention of "particles in the 0.2 - 0.6 $\mu m$" (line 404) intended to be a general reference to accumulation mode particles or a reference to particle sizes modelled here? If it is the latter, I am having trouble matching this with the numbers in Table 2 which gives a diameter of 0.2 $\mu m$ but the next size up is 2.2 $\mu m$.

Thanks for pointing this out, it is indeed the 0.2 $\mu m$ mineral dust particles lifetime we are referring to.

7. Nanometre and micrometre are used interchangeably (e.g. 0.2 $\mu m$ and 200 nm are both used). Some consistency referring to particles of the same size would be better.

A good suggestion, throughout the document $\mu m$ are now used.

8. In the comparison of the measured and model size distribution (lines 447-448), the authors say that the measured peak at 1 $\mu m$ matched the model well. The aged model

peak is at a little lower particle size distribution.

We agree the peak is a bit lower than the measured peak. Rewritten to:

*The measured size distribution of $^{137}Cs$ is bimodal with peaks around $1\,\mu m$ and $0.02\,\mu m$. The larger peak at $1\,\mu m$ fits well the released size distribution in FLEXPART. The peak of the aged size distribution dominated by particles of $0.6\,\mu m$.*

9. What are the upside down black triangles in Figure 5c? Are these the "daily median ratios" (line 455)?

see below pt. 11.

10. The mention of "aerosol type" on line 512 refers, I think, to splitting the aerosol size distribution up into bins with different particle sizes and scavenging rates. If this is correct, I find this terminology confusing since I would say that it is one aerosol type (i.e. 137Cs attached to sulphate) and not more than one aerosol type.

We would agree with your definition of type. "Type" was used here as it is not only size that can change, but also the properties of an aerosol particle. I.e BC can have very different hygroscopocity depending on the co-emitted gases and material (Huang et al., 2013). Though not applied in this study, it is possible to have separate types of BC particles (with different removal efficiencies) that both makes up "BC".Changed *"aerosol type"* to *"specific aerosol kind"*

11. Is the dotted black line in Figure 5d the 1-1 line or the line of best fit to the 133Xe (or both?)?

9 & 11 Alongside addition of FLEXPART version 9 result, an explanation of the upside down black triangles as the median daily station concentration values. Also that it is

the 1:1 line (black dotted) that is shown is added to Fig 5d, together with the log-fit to
FLEXPART v9 data from Kristiansen et al., 2015

12. I don't follow the final sentence of the paragraph on lines 559-561. I cannot see a
burden with a substantially different dependence on latitude in Figure 6. Furthermore,
simulations 5 and 8 also have a phase dependent change to the removal parameters.

You are perhaps correct in saying that a "substantially" changed latitudinal dependence
is not seen anywhere. The latitudinal column burden is dominated by the emission lat-
itude, thus it is not ideal to make too strong statements from this figure. Therefore
"substantially" has been replaced by "noticeable". A sentence was added to the dis-
cussion regarding surface concentrations of figure 7 where changes in the Arctic are
better displayed. The new sentence now reads:

*Only simulations #5 to #8, which have phase dependent changes to removal param-
eters, produce burdens with a noticeable different dependence on latitude when com-
pared to simulation #1.*

13. Lines 570-571. The authors refer to an increase in remote areas like the Arctic but
the increase seems much larger near the equator.

Thank you for pointing this out, you are absolutely correct. When describing this *"re-
mote areas like the Arctic"* has been changed to *"remote Tropical areas"*

14. It is difficult to see the black circles in Figure 7, particular since the coastlines are
also black

We changed markers to white and increased size somewhat.

**Supplement:**

[revised manuscript text omitted]